

# Five years of Aeolus wind profiling: global coverage and data quality

Oliver Lux[1], Michael Rennie[2], Jos de Kloe[3], and Oliver Reitebuch[1]

[1]Deutsches Zentrum für Luft- und Raumfahrt, Institut für Physik der Atmosphäre, Oberpfaffenhofen, Germany
[2]European Centre for Medium-Range Weather Forecasts, Research Department, Shinfield Park, Reading RG2 9AX, United Kingdom
[3]Royal Netherlands Meteorological Institute (KNMI), R&D Satellite Observations, Utrechtseweg 297, De Bilt 3731 GA, The Netherlands

**Correspondence:** Oliver Lux (oliver.lux@dlr.de)

**Abstract.**

The European Space Agency's Aeolus mission (2018–2023) was the first satellite to deliver global wind profile observations using Doppler wind lidar technology. Aeolus significantly advanced numerical weather prediction (NWP) and atmospheric science, particularly in improving forecast skill and understanding global circulation and aerosol transport. With its successor mission Aeolus-2 now in development, a detailed assessment of Aeolus' long-term performance is essential to guide future system design and processing strategies. This study analyses the evolution and interrelation of key parameters from Aeolus Level-1B (L1B) and Level-2B (L2B) data products from processor baseline 16, including signal-to-noise ratio (SNR), error estimate (EE), and wind data coverage. For the first time, L1B instrument parameters are interpolated onto the L2B wind grid, enabling direct correlation with final product quality and tracking of performance changes across the mission. A major focus is placed on wind data coverage. Traditional metrics counted valid observations without considering variable horizontal and vertical bin sizes. Here, we assess the atmospheric area covered, accounting for the varying horizontal and vertical extent of wind bins from the Rayleigh and Mie channel across different mission phases. This reveals important changes in data yield and the influence of events like wildfires (2019) and the Hunga Tonga eruption (2022). We also evaluate how well the EE represents actual wind uncertainty using observation minus short-range NWP forecast differences. Under low-SNR conditions, the Rayleigh-clear EE slightly overestimates random error, whereas in the high-SNR regime, the Mie-cloudy EE tends to underestimate uncertainty. These findings provide critical input for optimising Aeolus-2 instrument design and data processing and offer a valuable framework for future Doppler wind lidar missions by improving data evaluation, quality control, and assimilation readiness.

## 1 Introduction

The Aeolus mission, operated by the European Space Agency (ESA) from 2018 to 2023, was the first satellite to deliver global wind profile measurements using Doppler wind lidar technology (European Space Agency (ESA), 2008; Reitebuch, 2012; Reitebuch et al., 2020; Straume et al., 2020; Reitebuch, 2025). Over its nearly five-year lifetime, Aeolus significantly improved numerical weather prediction (NWP) (Rennie et al., 2021; Pourret et al., 2022; Martin et al., 2023; Halloran and Forsythe,





2024; Rennie and Isaksen, 2024; Shenolikar et al., 2025) and advanced scientific studies on atmospheric dynamics, aerosol
transport, and climate research (Wright et al., 2021; Lukens et al., 2022; Sun et al., 2024; Wang et al., 2024; Žagar et al., 2025;
Trapon et al., 2025). With the mission successfully completed, efforts are now focused on its successor, Aeolus-2, which will
build on Aeolus' achievements with key design modifications and enhanced data processing capabilities (Heliere et al., 2023;
EUMETSAT, 2023; Chalex et al., 2024; Flament et al., 2024; Ciapponi et al., 2024; Knobloch et al., 2025; Marksteiner et al.,
2025; Berceau et al., 2025). In parallel, extensive simulation activities are underway to refine the new instrument's performance
and optimise its operational setup.

A comprehensive assessment of the wind mission's long-term performance and data quality is essential for shaping Aeolus-2.
While past studies have primarily examined individual instrument components such as the laser (The Pierre Auger Collabo-
ration et al., 2024; Lux et al., 2020, 2024a), the Rayleigh and Mie spectrometer channels (Witschas et al., 2022; Vaughan
et al., 2025), and detectors (Weiler et al., 2021a; Lux et al., 2024b, 2025), a broader analysis of the evolution of Aeolus data
product quality has been missing. The recent reprocessing of Aeolus data, with Baseline 16 (B16) as the first consistent dataset
covering the entire mission period, now enables a detailed investigation of key data product parameters, including wind speed,
error estimates, and signal-to-noise ratio (SNR).

Up to now, the amount of valid wind data contained in the Aeolus Level-2B product has typically been assessed by counting
the number of observations, without considering the spatial extent of each wind bin (Rennie et al., 2021). However, for Aeolus,
both the horizontal and vertical bin sizes vary for the two receiver channels and depended on the selected processor settings.
The Rayleigh channel generally used long horizontal integration lengths of about 86 km for deriving the wind speed from
clear-air regions, while the Mie channel initially operated with much shorter bins of around 10-20 km, targeting aerosol and
cloud layers. Throughout the mission, changes in signal strength–particularly due to the progressive decline in atmospheric
return–required adjustments in onboard accumulation, introducing further variability in horizontal bin sizes. Vertically, bin
thickness ranged from 250 m to 2 km, differing not only between the Rayleigh and Mie channel but also across latitudinal
bands and mission phases. As a result, the conventional count-based approach underrepresents the actual atmospheric volume
sampled by Aeolus. To provide a more accurate picture of Aeolus' observational reach and data yield, this study presents a
detailed analysis of area coverage, accounting for the variable dimensions of wind bins under different conditions during the
mission lifetime.

Understanding the performance of the Rayleigh and Mie channels throughout different mission phases, under both high and
low atmospheric backscatter conditions, and in the presence of phenomena like stratospheric aerosols is particularly important
for anticipating the capabilities of Aeolus-2. However, directly correlating instrument parameters that are provided in the Level-
1B (L1B) product, such as SNR, with key Level-2B (L2B) output parameters used in NWP and scientific research, such as
the wind speed and its assigned estimate of the wind error standard deviation, is complicated by differences in the gridding
between the two data levels. To overcome this, we averaged L1B data onto the L2B grid, enabling a consistent framework to
analyse the temporal evolution of critical performance parameters. This approach allows for a more accurate assessment of
their influence on wind data quality and supports the refinement of retrieval algorithms through optimised parameter thresholds
and quality control procedures.



This study presents a detailed analysis of the evolution of the most relevant Aeolus L1B and L2B parameters, their interdependencies, and the overall wind data coverage throughout the mission. The results will provide valuable insights for optimising Aeolus-2 data processing and ensuring the next generation of spaceborne wind observations continues to enhance weather prediction and atmospheric research.

The article is organised as follows. Section 2 introduces the methods and datasets used in the analysis, beginning with a brief description of the design and operating principle of the Aeolus Doppler wind lidar (Sect. 2.1). This is followed by an overview of the Aeolus data structure (Sect. 2.2), including the Level-1 (Sect. 2.2.1) and Level-2B (Sect. 2.2.2) products, as well as the methodology for linking parameters across the two levels (Sect. 2.2.3). Additional information on the Level-2B processor is provided in Appendix A. Section 2.3 presents an overview of instrument performance over the mission lifetime, while Sect. 2.4 discusses the different horizontal integration lengths applied throughout the mission.

The results are presented in two main parts. Section 3.1 focuses on wind data coverage, examining its temporal evolution (Sect. 3.1.1), latitudinal dependence (Sect. 3.1.2), and sensitivity to range bin settings (Sect. 3.1.3). Section 3.1.4 provides a detailed view of Mie-cloudy wind coverage and its global distribution across different mission phases. The second part of the results, presented in Sect. 3.2, explores the SNR and its connection to wind observation errors. This includes a discussion of wind error estimates (Sect. 3.2.1) and their comparison with errors relative to ECMWF model background winds (Sect. 3.2.2). Definitions of the SNR and related quantities used in this study are given in Appendix B. The article concludes with a summary and outlook, highlighting implications for the follow-on mission Aeolus-2.

## 2 Methods and Datasets

This section provides a brief overview of the ALADIN Doppler wind lidar instrument design and its measurement principle, which are essential for understanding the key parameters defined later in the text and for interpreting the factors that influenced instrument performance throughout the mission. The structure of the Aeolus data products is then introduced, with a focus on the transition from the Level-1 to the Level-2 data products. While the basic processing steps are outlined here, a more detailed description is provided in Appendix A. Finally, the section summarises the instrument's performance over its nearly five-year mission lifetime and presents an overview of the datasets analysed in this study to assess wind data quality at various stages of the mission.

### 2.1 Design and operation principle

The Atmospheric LAser Doppler INstrument (ALADIN) was the sole payload of the Aeolus satellite, which operated in a sun-synchronous orbit at an altitude of approximately 320 km, completing a full cycle every seven days. The instrument comprised a pulsed, frequency-stabilised, ultraviolet (UV) laser transmitter, transmit-receive optics (TRO), a Cassegrain-type telescope, and a dual-channel receiver capable of detecting backscattered signals from both atmospheric particles and molecules using high-spectral resolution lidar (HSRL) technique (European Space Agency (ESA), 2008; Reitebuch, 2012).





ALADIN measured wind velocities by analysing the frequency shift between emitted and backscattered laser pulses, induced by the Doppler effect. As the laser beam interacted with molecules and aerosols moving along the laser line-of-sight (LOS) with velocity $v_{\mathrm{LOS}}$, the frequency of the backscattered signal was shifted according to $\Delta f = \frac{2f_0 v_{\mathrm{LOS}}}{c}$, where $c$ is the speed of light, and $f_0$ is the frequency of the emitted light. To gain sensitivity to the horizontal wind component along its LOS, the telescope was tilted $35°$ off-nadir, corresponding to approximately $37.5°$ at the surface due to Earth's curvature.

A schematic representation of ALADIN is provided in Fig. 1. The instrument is equipped with two redundant laser transmitters, Flight Models (FMs) A and B, that can be alternated using a flip-flop mechanism (FFM). These transmitters are diode-pumped, injection-seeded Nd:YAG lasers operating in a master oscillator power amplifier configuration, followed by a frequency tripling stage to generate frequency-stable, narrowband UV radiation at 354.8 nm. For a more detailed discussion on the laser transmitter design, performance, and development challenges, see Lux et al. (2024a) and references therein.

Before exiting the laser system, a small fraction (0.5%) of the UV light is extracted at a beam splitter (BS) within the TRO configuration. After attenuation, this portion is guided to the instrument's field stop and receiver channels. Referred to as the internal reference path signal, it is used to determine the frequency of the outgoing beam and calibrate the frequency-dependent transmission of the receiver spectrometers. As the laser beam propagates through the TRO, it passes through a quarter-wave plate (QWP), which ensures circular polarisation. This polarisation state enables reflection of the return signal toward the receiver, as the instrument operates in a monostatic configuration, using the same telescope for both signal emission and reception.

The backscatter signal from the atmosphere and the ground is collected by the telescope and directed toward the optical receiver through a field stop with an 88 μm diameter, limiting the receiver's field of view to 18 μrad. This design reduces solar background noise but requires precise pointing stability due to the receiver spectrometers' high angular sensitivity. ALADIN's receiver comprises two complementary channels to derive the Doppler frequency shift from distinct scattering mechanisms: a Mie channel for detecting narrowband backscatter from clouds and aerosols (Vaughan et al., 2025), and a Rayleigh channel for broadband molecular backscatter (Witschas et al., 2022). The Mie channel is implemented using a Fizeau interferometer and relies on the fringe-imaging technique (McKay, 2002), where the Doppler frequency shift is derived from the spatial displacement of the fringe centroid relative to that of the internal reference signal.



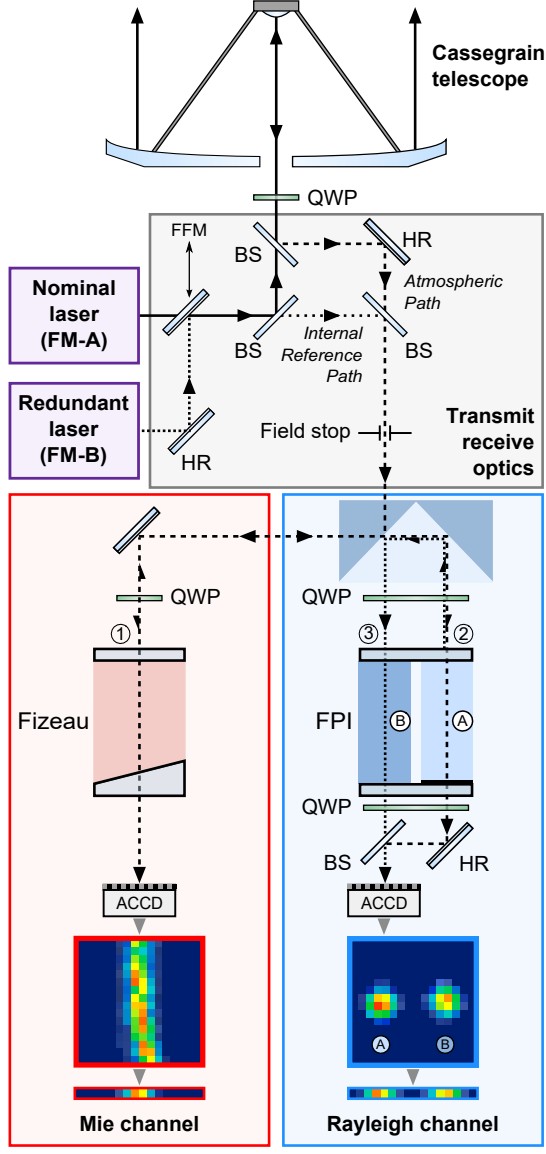

**Figure 1.** Simplified schematic of the ALADIN instrument onboard Aeolus comprising two switchable UV laser transmitters, a Cassegrain telescope, transmit-receive optics, and a dual-channel receiver. Note: HR – highly-reflective mirror; FFM – flip-flop mechanism; BS – beam splitter; QWP – quarter-wave plate; FPI – Fabry–Pérot interferometer; ACCD – accumulation charge-coupled device.

The Doppler shift from the broadband Rayleigh–Brillouin backscatter spectrum is derived using two sequential Fabry–Pérot interferometers (FPIs) that implement the double-edge technique (Flesia and Korb, 1999). Acting as bandpass filters of suitable width and spacing, the FPIs are symmetrically placed around the laser frequency, and the frequency shift is obtained from the contrast between the two transmitted signals.



The atmospheric signal – comprising the Fizeau interferometer fringe and the two circular spots from the FPIs – is recorded
with accumulation charge-coupled device (ACCD) detectors, whose principle and performance are described in Lux et al.
(2024b). Charges are collected in a 16 × 16 pixel array of the ACCD image zone (see Fig. 1) and, during wind measurement
mode, vertically binned into a single row before transfer to the ACCD memory zone, which stores up to 25 rows (one per
vertical range bin). Of these, one range bin is reserved for the solar background signal, which is subtracted from the atmospheric
backscatter. To improve SNR, signals from a configurable number of successive laser pulses are accumulated onboard prior to
readout. The accumulation settings varied during the mission, as detailed in the next section.

## 2.2 Aeolus data structure

Aeolus data products are categorised into Level 0 (L0), Level 1 (L1), and Level 2 (L2), based on processing level. The pro-
cessing starts with raw telemetry data, downlinked as Annotated Instrument Source Packets (AISP), which are time-ordered
and reformatted according to ESA's Ground Segment File Format Standard (European Space Agency (ESA), 2003) to create
the Level 0 product. This product includes information from the Attitude and Orbit Control System (AOCS), such as time,
position, velocity, and attitude, along with housekeeping data (e.g., laser frequency, energy, and instrument mode), internal
reference data from the Rayleigh and Mie channels, and detector readout signals from the two receiver channels.

In L1, atmospheric return signals are structured into *observations*, each spanning 12 seconds and corresponding to a horizon-
tal resolution of 86 km. Each observation comprises multiple *measurements*, with the number of measurements (N) determined
by the laser pulses (P) over which signals are accumulated onboard. At the beginning of the mission, P was set to 20 and N to
30, with one pulse lost during detector readout (P–1 setting). From February 2019 onward, P was reduced to 19, resulting in
18 accumulated pulses per measurement. To improve signal levels relative to readout noise, the P/N settings were adjusted in
December 2021 and again in April 2022, first to 38/15 and later to 114/5 (Lux et al., 2024b). Consequently, the total number
of laser pulses per observation, calculated as (P–1)·N, increased from 540 initially to 555 and finally to 565 in later mission
phases.

### 2.2.1 The Level-1A and Level-1B product

The next stage of the Aeolus processing chain refines AOCS and housekeeping data from L0 to derive geolocation information
for each measurement in the Level-1A (L1A) product. Additionally, recorded data is categorised by instrument mode, such
as response calibration or wind measurement mode. The L1A processor iterates over different instrument modes, processes
packets into annotation and measurement datasets, and generates mode-specific L1A product files.

The L1A wind measurement data (ALD_U_N_1A) provides geolocated signal levels per observation, measurement, and
atmospheric range bin. Additionally, Rayleigh and Mie signal levels are recorded for each of the 20 pixels of the respective
ACCD. Of these, the first two and last two pixels are unilluminated and used for quantifying the detection chain offset (DCO).
These virtual pixels, which results from a special clocking sequence during readout, contain only the minimal charges generated
in the register during readout, enabling DCO correction for the 16 active pixels per detector array row (Weiler et al., 2021b;
Lux et al., 2024b).





The L1B product provides pre-processed, quality-annotated atmospheric wind data, acting as an intermediate step between raw measurements and higher-level wind products. It includes geolocation data, confidence metrics, calibration parameters, and pre-processed Mie and Rayleigh measurement information. Key components include Rayleigh and Mie signal intensity (useful signal) for each observation, measurement, and atmospheric range bin, as well as wind velocity for both channels on the same grid (measurement x range bin). While the wind velocity data is derived from the spectrometer responses using calibration mode outputs, it has not yet been corrected for atmospheric temperature and pressure influencing its spectral width, which is an essential step for accurately deriving winds from the Rayleigh spectrometer. Moreover, Mie contamination of the Rayleigh signals has not yet been accounted for. The Rayleigh winds are therefore not useful for assimilation in NWP models or scientific applications as it comprises atmospheric scene-dependent biases. Additionally, the L1B product provides critical input parameters for L2 processing, including the SNR and the scattering ratio (SR), defined as the ratio of total backscatter (molecular plus particulate) to molecular backscatter.

Beyond wind velocity mode, ALADIN operated in several other instrument modes throughout the mission for characterisation, monitoring, and calibration purposes. These modes require distinct processing within the L1B processor, as L2 processors are designed exclusively for deriving the Aeolus wind (L2B) and aerosol (L2A) product (Flament et al., 2021). Further details on the L1B processor can be found in Reitebuch et al. (2018).

### 2.2.2 The Level-2B product

The Level-2B (L2B) wind data product is the primary output of the Aeolus mission, as it provides the horizontal line-of-sight (HLOS) wind speed derived from the Rayleigh and Mie channels. These wind measurements are suitable for assimilation into NWP systems and for scientific research. To generate these data, the L2B processor uses L1B inputs and applies an advanced retrieval algorithm that incorporates multiple corrections and accounts for various error sources. In particular, the calibrated Rayleigh wind measurements from the L1B product are corrected for atmospheric temperature and pressure effects. Moreover, Rayleigh and Mie winds are corrected for orbital variations in the spectrometer incidence angle, which are correlated with temperature gradients of the primary telescope mirror (Weiler et al., 2021b; Rennie et al., 2021).

In addition, the L2B processor performs a scene classification based on L1B parameters to distinguish between measurements made in regions with low aerosol or cloud content (*clear*) and those with sufficient aerosol or cloud presence (*cloudy*). These are then grouped to form Rayleigh-clear and Mie-cloudy wind results. The grouping length is controlled by L2B processor settings and reflects a trade-off between spatial resolution and noise. A more detailed description of the L2B processing algorithm, including the grouping algorithm and scene classification methodology, is provided in Appendix A.

L2B wind data and associated parameters, such as geolocation, error estimates, and validity flags, are provided as one-dimensional arrays. This contrasts with the L1B data format, which is organised on a two-dimensional grid (measurement × range bin). The horizontal extent of each L2B wind result depends on the number of L1B measurements that were grouped together during processing. In Baseline 16, the majority of Rayleigh-clear winds are derived from all, or nearly all, of the N measurements within a 12-second observation. In contrast, Mie-cloudy wind bins were typically set to a horizontal length of approximately 10 km. A quantitative overview of bin lengths used during different mission phases is provided in Sect. 2.4.



### 2.2.3 Correlation between L1B and L2B parameters

Due to the different gridding schemes of the L1B and L2B products (see Fig. A2), the relationships between relevant parameters from both products are often overlooked. Even within the Aeolus data calibration and validation community, analyses tend to focus on parameters from one product, most commonly the L2B HLOS wind results and their associated error estimates.

However, for processor refinements and the development of new algorithms for reprocessing of Aeolus products and simulations for Aeolus-2, it is valuable to investigate correlations between key L1B and L2B parameters. To enable such analysis, harmonisation of the two datasets is necessary in order to associate each L2B wind result with its contributing measurement bins.

In this study, harmonisation is achieved by computing average values over all measurement bins that contribute to each

wind result. The primary parameter considered is the L1B SNR from the Rayleigh and Mie channel. Here, it is deemed more meaningful to calculate the root-mean-square (RMS) value rather than the arithmetic mean, as the RMS gives more weight to higher values within a sample. This approach better represents the influence of high-SNR measurements, which is appropriate since these measurements contribute more significantly to the wind retrieval process, where the signals from all contributing bins are summed prior to deriving the Doppler frequency shift. Consequently, the SNR averaged over all $n$ contributing L1B

measurement bins is given as:

$$
\mathrm{SNR_{RMS}} = \sqrt{\frac{1}{n} \sum_{i=1}^{n} \mathrm{SNR}_i^2} \ . \tag{1}
$$

The definitions of the Rayleigh and Mie SNR are introduced in Appendix B.

### 2.3 Mission performance and dataset selection

This study aims to provide an overview of how the most relevant Aeolus L1B and L2B parameters evolved over the mission

period, which lasted from September 2018 through April 2023. During this time, instrument performance varied significantly. A comprehensive analysis of the performance evolution is provided in the Final Report of The Aeolus Data Innovation and Science Cluster (DISC) (2024); therefore, only a brief summary is presented here.

Following the launch of Aeolus in 2018, the primary laser (FM-A) operated with an initial UV output energy of 65 mJ. However, it exhibited a steady decline of about -1 mJ per week, falling below 40 mJ by mid-2019. This loss in laser energy

reduced signal levels along the instrument's optical path and led to correspondingly higher wind random errors (Lux et al., 2020). In June 2019, operations were switched to the redundant FM-B laser, which started with a higher initial UV output of 67 mJ and showed a slower degradation rate. Subsequent laser tuning increased its output to over 100 mJ; however, atmospheric signal levels decreased by more than 70% over the following three years. Unlike during the FM-A phase, this signal loss originated not within the laser itself, but along the FM-B-specific optical path (The Pierre Auger Collaboration

et al., 2024)–most likely due to laser-induced contamination in the relay optics in the TRO, including the FFM arrangement (Fig. 1).





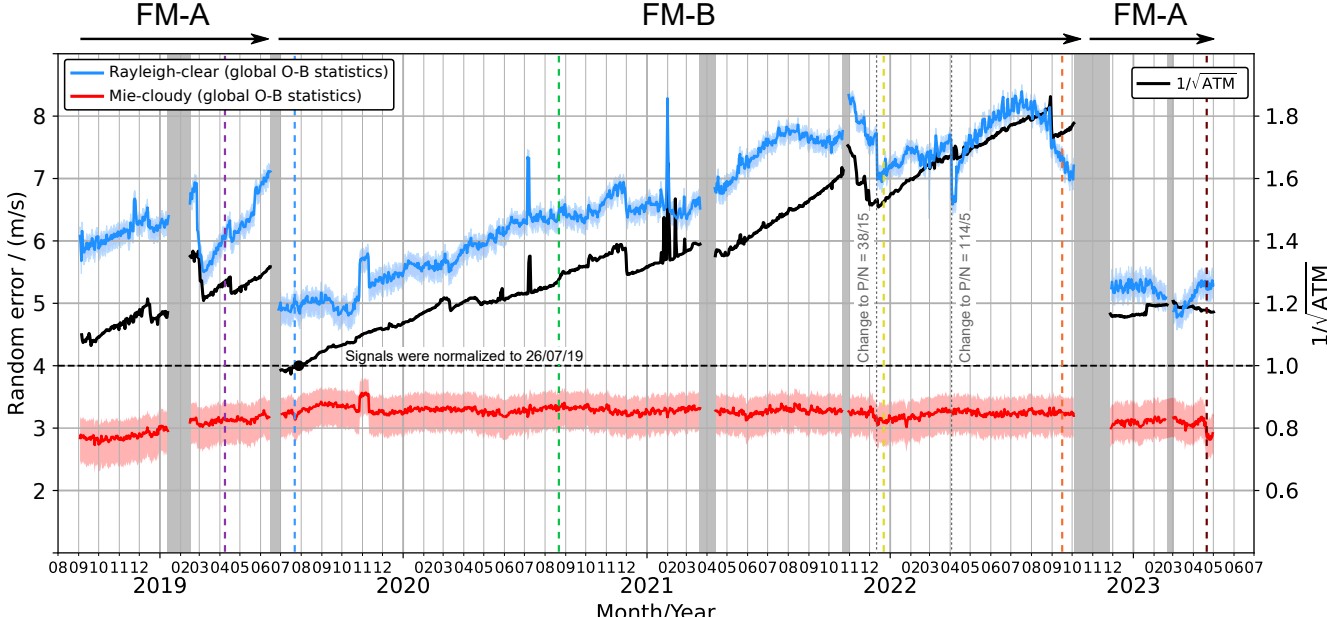

**Figure 2.** Timeline of the global, daily average random error for Aeolus Rayleigh-clear (blue) and Mie-cloudy (red) wind observations over the mission period, derived from (O–B) statistics assuming a model background error of 2.0 m s$^{-1}$. Shaded areas indicate the uncertainty range corresponding to assumed model background errors of 1.5 m s$^{-1}$ (upper bound) and 2.5 m s$^{-1}$ (lower bound). The black line (right axis) shows the reciprocal square root of the atmospheric signal level ($1/\sqrt{\text{ATM}}$), normalised to its value on 26 July 2019. Grey-shaded regions indicate periods when the instrument was temporarily switched off. Coloured vertical dashed lines mark specific days selected for analysis in this study.

In addition to increasing the laser energy, the number of accumulated pulses per measurement was raised from 18 to 38 in December 2021, and further to 114 in April 2022, as noted earlier. Consequently, the number of measurements per observation was reduced from 30 to 15, and later from 15 to 5. This approach helped to mitigate the impact of read-out noise and DCO noise on the overall Rayleigh noise, which had increased due to declining atmospheric signal levels, thereby improving the Rayleigh-clear random error. In November 2022, the mission reverted to the FM-A laser to restore transmission in the TRO. Despite its lower energy output of around 50 mJ, this second FM-A phase yielded atmospheric signals more than twice as strong as those at the end of the 40-month FM-B period. Benefiting from improved thermal management and alignment strategies for the laser developed during FM-B operations, this final phase maintained stable laser performance until the end of nominal operations in April 2023. Afterwards, a series of special tests, referred to as end-of-life activities, was performed to address a number of instrument-related and scientific questions until the final switch-off of the instrument on 5 July 2023 (Lux et al., 2024a).

The temporal evolution of Rayleigh and Mie random errors over the entire mission period, derived from differences between the HLOS wind observation and the ECMWF model short-range (up to 12 hour) forecast equivalent, referred to as observation-minus-background (O-B) departure statistics (Rennie and Isaksen, 2024), is shown in Fig. 2. The blue and red areas represent




daily averages from global wind data over all altitudes extracted from the L2B product (Baselines 16), while assuming a model background error of $(2.0 \pm 0.5)$ m s$^{-1}$. The short-range forecast (background) noise does vary spatially, e.g., being largest in the tropical upper-troposphere and lower stratosphere, but this range is typical for the ECMWF model, as determined via comparisons to radiosonde winds. The O-B departures are derived from data assimilation experiments in which Aeolus was not assimilated, but passively monitored. The instrument was switched off several times, either following Failure Detection Isolation and Recovery (FDIR) events or to switch between the two lasers, resulting in data gaps.

The black line illustrates the long-term trend of the reciprocal square root of the atmospheric signal level (ATM), as measured by the Rayleigh channel under clear-sky conditions (SR < 1.3) at approximately 10 km altitude. This value, $1/\sqrt{\text{ATM}}$, is normalised to the beginning of the FM-B period in July 2019. Signal levels dropped by about 70% between that time and September 2022, i.e., $1/\sqrt{\text{ATM}}$ increased from 1.0 to $1/\sqrt{0.3} \approx 1.8$. Temporary increases in the signal can be attributed to specific laser operations to boost the output energy, such as those in December 2020 and November 2021.

The evolution of the Rayleigh-clear random error broadly follows the trend of $1/\sqrt{\text{ATM}}$, as it is primarily driven by the SNR (see Appendix B), with Poisson noise being the dominant contributor (Reitebuch et al., 2018). Seasonal modulations are caused by variations in solar background noise, which adds to the shot noise. This effect is strongest during boreal summer, with secondary maxima during austral summer.

In contrast, the evolution of the Mie-cloudy random error is more strongly affected by data processing algorithms and configuration changes than by the signal trend. One notable example is the application of a dedicated range bin setting in October and November 2019, aimed at investigating the correspondence between Aeolus observations and wind measurements derived from Atmospheric Motion Vectors (AMVs).

To capture the different mission phases, six days of data from early 2019 to early 2023 were selected for analysis (see also coloured vertical lines in Fig. 2). The corresponding orbit numbers, number of L1B observations, and the active laser transmitter are listed in Table 1, along with the atmospheric signal level (ATM) and the P/N setting.

**Table 1.** Aeolus one-day datasets used in this study, including the corresponding orbit numbers, the active laser during each period, the atmospheric path signal level relative to 26 July 2019, and the applied P/N settings. Note that only (P–1) pulses were accumulated in the processing.

| Dataset | Day | Orbit numbers | Number of observations | Active laser | ATM signal level | P/N setting |
|---|---|---|---|---|---|---|
| 1 | 08/04/2019 | 03617 – 03632 | 6761 | FM-A | 0.61 | 19/30 |
| 2 | 20/07/2019 | 05251 – 05266 | 7205 | FM-B | 1.00 | 19/30 |
| 3 | 21/08/2020 | 11562 – 11577 | 6884 | FM-B | 0.61 | 19/30 |
| 4 | 18/12/2021 | 19443 – 19457 | 7128 | FM-B | 0.43 | 38/15 |
| 5 | 15/09/2022 | 23534 – 23549 | 7115 | FM-B | 0.33 | 114/5 |
| 6 | 25/04/2023 | 27054 – 27069 | 6936 | FM-A | 0.72 | 114/5 |





## 2.4 Horizontal bin length of the L2B wind results

The changes in P/N settings during the mission led to variations in the accumulation length, i.e., horizontal bin length, when grouping L1B measurements into L2B wind results. Figure 3 shows the number of measurements contributing to each Mie-cloudy (a) and Rayleigh-clear (b) wind result. In the early mission phase with N = 30, nearly half of the Mie-cloudy winds were derived from fewer than three measurements, corresponding to a horizontal bin length of approximately 9 km. During the FM-B phase, also at N = 30, about 80 % of the Mie-cloudy winds were based on four measurements (≈11 km). With the change to N = 15, 90 % of Mie-cloudy winds were formed from three measurements (≈17 km). Finally, with the N = 5 setting introduced in April 2022, all Mie-cloudy wind results were based on a single measurement, maintaining a horizontal bin length of approximately 17 km.

For Rayleigh-clear winds, the vast majority (around 80 %) of results were based on all N measurements, corresponding to a horizontal bin length of 86 km. Even among the remaining 20 %, more than half of the available L1B measurements were typically used to form each wind result, regardless of the P/N setting.

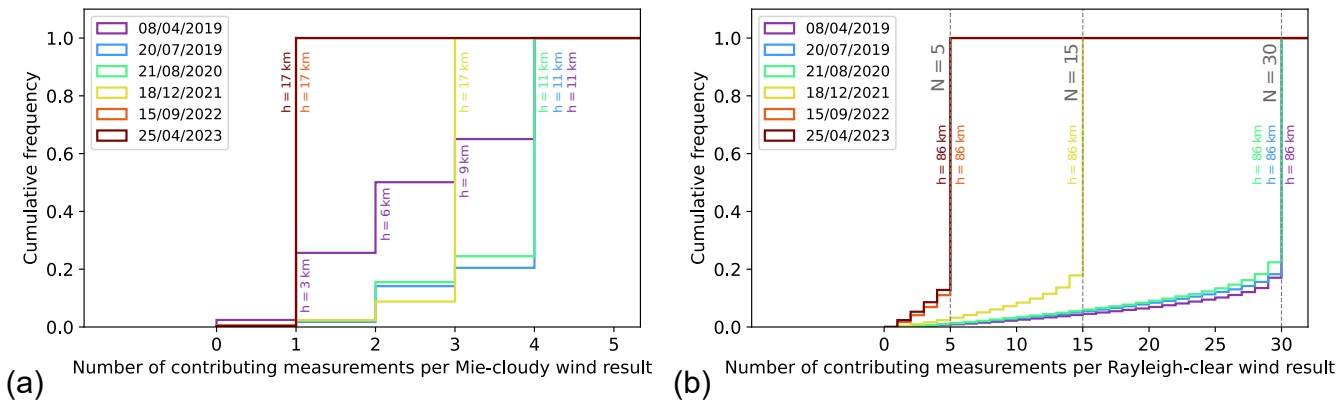

**Figure 3.** Histograms showing the number of L1B measurements contributing to (a) L2B Mie-cloudy and (b) L2B Rayleigh-clear wind results for six selected days throughout the mission period. The coloured labels denote the horizontal bin lengths $h$ that correspond to the number of grouped measurements depending on the P/N settings, as listed in Table 1.

## 3 Evolution of wind data quality throughout the Aeolus mission

Regarding systematic error, Aeolus wind data quality was primarily affected by the increasing number of hot pixels (Weiler et al., 2021a; Lux et al., 2024b, 2025) and by orbital temperature variations of the telescope's primary mirror (M1) (Weiler et al., 2021b). These effects were largely mitigated by regular dark current measurements and the implementation of an M1 bias correction scheme. Reprocessing campaigns during and after the operational mission phase further reduced biases in both channels (The Aeolus Data Innovation and Science Cluster (DISC), 2024), achieving stable values that met mission requirements, i.e., below $0.7 \, \mathrm{m \, s^{-1}}$ (European Space Agency (ESA), 2016).





The random errors of Mie-cloudy and Rayleigh-clear winds, however, showed greater variability over the mission period (see Fig. 2), mainly driven by the atmospheric backscatter signal and changes in the P/N settings (Table 1). In addition to the random error, typically quantified as the standard deviation, $\sigma$, from global (O–B) statistics (Rennie et al., 2021; Rennie and Isaksen, 2024), it is also valuable to analyse the geographical and vertical distribution of wind observations. To this end, the spatial

and vertical coverage of the wind data was assessed, as described in the following section, and related to the evolution of Mie and Rayleigh random errors across different mission phases and atmospheric conditions, such as the presence of stratospheric aerosols following the Hunga Tonga volcanic eruption in 2022.

In the next step, the wind errors were correlated with the Rayleigh and Mie SNR values, enabling an investigation of the relationship between data coverage, wind error, and SNR. Furthermore, the estimate of the wind error standard deviation,

provided in the L2B product for each wind result, were analysed and compared with the actual wind error relative to the model background. This comparison offers valuable insights into the representativeness of the provided error estimates and reveals performance limitations encountered during the Aeolus mission–limitations that must be addressed in preparation for Aeolus-2.

## 3.1 Wind data coverage

Data coverage of L2B Rayleigh-clear and Mie-cloudy wind results is evaluated for selected days during the mission period,

as listed in Table 1. It is defined as the ratio of the area containing useful wind data to a reference area. The reference area (i.e., full coverage) corresponds to the total area spanned by all Rayleigh-clear or Mie-cloudy wind results–regardless of validity–extending from the ground up to the top edge of the uppermost range bin of the respective wind product. Useful wind data are defined as those L2B results flagged as valid and whose wind departure $x_i = v_{i,\text{L2B}} - v_{i,\text{ECMWF}}$ with respect to the ECMWF background model does not exceed a modified Z score of 3.5.

The modified Z score $Z_\mathrm{m}$ is a robust method for identifying and removing gross errors, particularly suitable for datasets that are small or non-normally distributed. In the context of Aeolus wind data, it was shown to be effective for filtering out outliers, especially since the error estimates provided in the L2B product are not sufficiently accurate (Lux et al., 2022). The modified Z score is computed as the distance from the median, normalised by the scaled median absolute deviation (MAD), and is given by Iglewicz and Hoaglin (1993):

$$Z_{\mathrm{m},i} = \frac{x_i - \mathrm{median}(x_i)}{\text{scaled MAD}} \quad \text{with} \quad \text{scaled MAD} = 1.4826 \cdot \mathrm{median}(|x_i - \mathrm{median}(x_i)|) \tag{2}$$

Since the median and scaled MAD are less sensitive to outliers than the mean and standard deviation, this approach provides a more robust filtering of unreliable data. According to Iglewicz and Hoaglin (1993), observations with an absolute modified Z score exceeding 3.5 should be considered outliers. This filtering is particularly important in low-SNR regions, such as those below clouds. In Baseline 16, a significant portion of such winds (a few percent) are flagged as valid despite their large wind

error, but are identified based on the modified Z score criterion, thereby ensuring that the reported coverage reflects truly usable wind measurements.




Figure 4 depicts the valid and quality-controlled ($Z_\mathrm{m} < 3.5$) Rayleigh-clear and Mie-cloudy wind results from a selected orbit on 20 July 2019. Grey areas indicate wind data that are either flagged as invalid or have a modified Z score above 3.5. For this particular orbit, 3.4 % of the valid Rayleigh-clear data and 2.0 % of the valid Mie-cloudy data are discarded by the modified Z score filter. Data coverage is determined per 1 km altitude interval by calculating the ratio of the area containing useful wind results to the total area spanned over that altitude range. This procedure is illustrated by the blue and red horizontal bars in Fig. 4. The outlined areas in panels (b) and (d) represent the useful wind data, whereas the grey areas correspond to invalid data or gross errors.

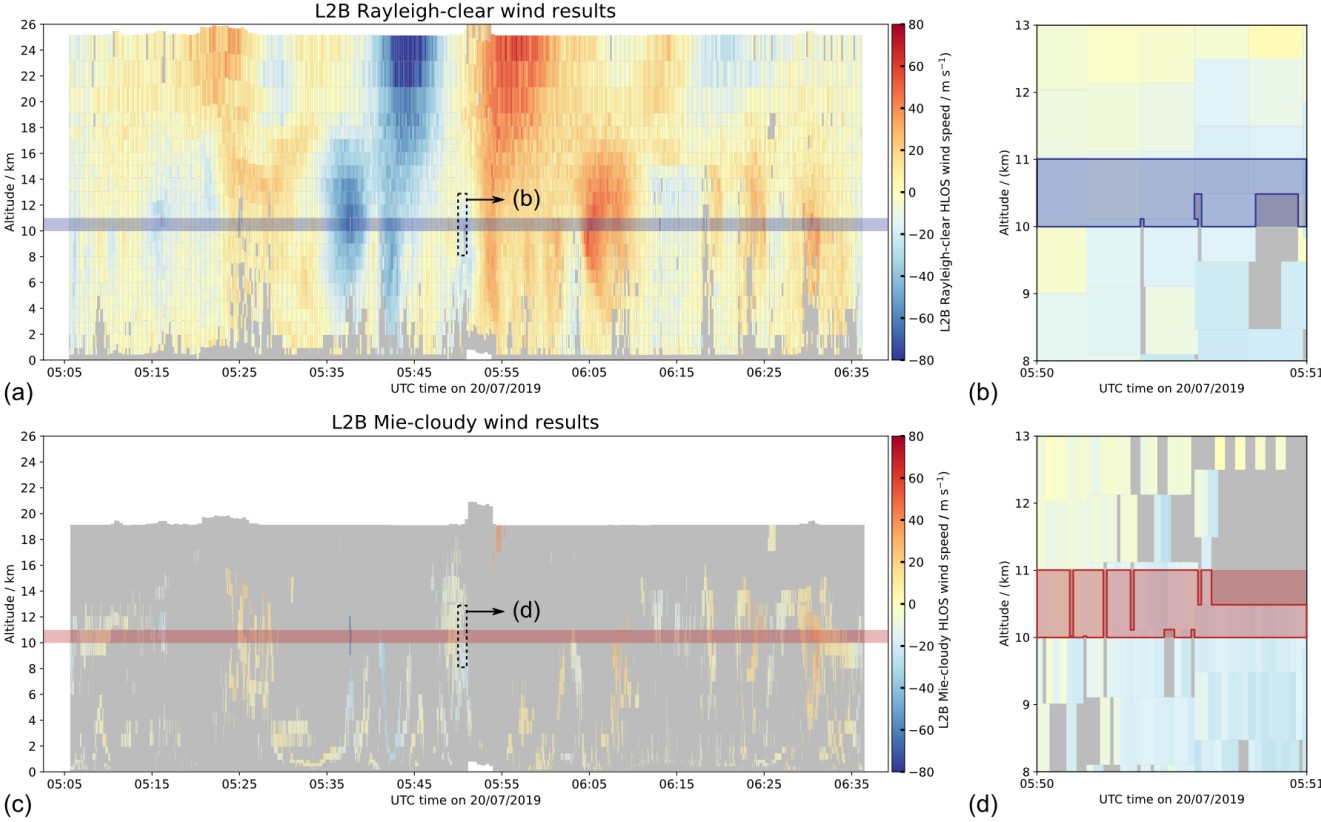

**Figure 4.** a, b) Aeolus L2B Rayleigh-clear and (c, d) Mie-cloudy HLOS wind results measured on 20 July 2019 between 05:05 UTC and 06:36 UTC (orbit number 5254). The grey-shaded area indicates invalid data. Together with the coloured valid wind data, it shows the overall coverage of the L2B data grid for both channels. Coverage is determined per 1 km altitude range by calculating the ratio of the area containing wind results–outlined in blue and red in panels (b) and (d)–to the total area spanned over that altitude range.

In the next step, the wind data are classified according to the HLOS wind speed difference relative to the ECMWF model background, assuming a model background standard error of 2.5 m s$^{-1}$. This absolute value of the (O-B) departure is referred to as $\varepsilon$ in the following. Wind results with $\varepsilon < 2.5$ m s$^{-1}$ are regarded as "high-quality", those with 2.5 m s$^{-1} < \varepsilon < 5.0$ m s$^{-1}$





are classified as "medium-quality", and results with $\varepsilon > 5.0\ \mathrm{m\,s^{-1}}$ are considered "low-quality". The limit for high-quality winds was chosen in accordance with the mission requirement for the wind random error in the free troposphere from 2 to 16 km (European Space Agency (ESA), 2016). For altitudes below 2 km, a stricter requirement of $1\ \mathrm{m\,s^{-1}}$ is formulated in

the requirements, due to the availability of Mie-cloudy winds in the planetary boundary layer (PBL), which generally exhibit lower random errors than Rayleigh-clear winds.

The $5.0\ \mathrm{m\,s^{-1}}$ threshold between medium- and low-quality winds follows the requirements specified in the Observing Systems Capability Analysis and Review (OSCAR) database of the World Meteorological Organisation (WMO) for global NWP and real-time monitoring in the upper troposphere and lower stratosphere as well as in the PBL (WMO, 2025).

It should be noted that the assumption of a constant short-range forecast random error is a simplification to support the classification of wind results into high-, medium-, and low-quality via the $\varepsilon$ parameter. In reality, the short-range forecast errors are situation-dependent (flow-dependent) and with temporal averaging vary with location, altitude, forecast time, and other factors. For this study, we adopt a relatively high value of $2.5\ \mathrm{m\,s^{-1}}$ to represent a best-case scenario for the Aeolus random error, thereby maximising the coverage of high-quality winds. The shaded areas in Fig. 2 represent the the uncertainty

range of the Aeolus wind random error corresponding to assumed background forecast errors of $1.5\ \mathrm{m\,s^{-1}}$ (upper bound) and $2.5\ \mathrm{m\,s^{-1}}$ (lower bound). This illustrates that, due to the larger observation error, the Rayleigh-clear random error is only slightly influenced ($\approx \pm\,0.1\ \mathrm{m\,s^{-1}}$) by the choice of assumed model error within the range from 1.5 to $2.5\ \mathrm{m\,s^{-1}}$, whereas the uncertainty is larger for the Mie-cloudy winds ($\approx \pm\,0.3\ \mathrm{m\,s^{-1}}$).

Another important contribution is the representativeness error, which in NWP data assimilation refers to the uncertainty

arising from the mismatch between what an observation actually measures and what the forecast model is assumed to represent. Three main sources can be distinguished:

- scale mismatch, since observations capture atmospheric conditions at a point or over a small area, whereas NWP models represent averages over grid cells that can span tens of kilometres;

- forward-modelling errors, for example in Aeolus where model winds are converted to HLOS winds assuming a point

wind rather than averaging over the range bin weighted by attenuated backscatter;

- sub-grid processes, such as convective clouds or microphysical phenomena that models cannot adequately resolve due to limitations of parameterised physics.

The magnitude of representativeness error is difficult to quantify, but it is expected to be larger for Mie-cloudy winds than for Rayleigh-clear winds, as models cannot perfectly capture cloud processes and associated dynamics due to limited resolution

and imperfect parameterisations. An estimate of the representativeness error for Mie-cloudy winds is provided in Sect. 3.2.2, based on the correlation between the (O–B) wind speed difference and the error estimate from the L2B product.





### 3.1.1 Evolution of the global wind data coverage

The area coverage of Mie-cloudy and Rayleigh-clear winds as a function of altitude is shown in Fig. 5. Each of the six panels corresponds to a selected day of global wind data, and the three red (Mie) and blue (Rayleigh) colour shades represent the

defined quality classes (high, medium and low), as described above. Several noteworthy features emerge from the plots. The coverage of Rayleigh-clear wind data exhibits a characteristic altitude dependence, with a maximum around 10 km and decreasing coverage toward both lower and higher altitudes. This pattern closely follows the altitude dependence of the Rayleigh SNR (see Sect. 3.2), which is directly influenced by the atmospheric backscatter signal. While molecular density, and thus backscatter strength, increases toward the surface, this effect is offset by greater range and increased atmospheric extinction,

also from increased cloud coverage, at lower altitudes, resulting in a peak SNR (and therefore coverage) between 6 and 10 km.

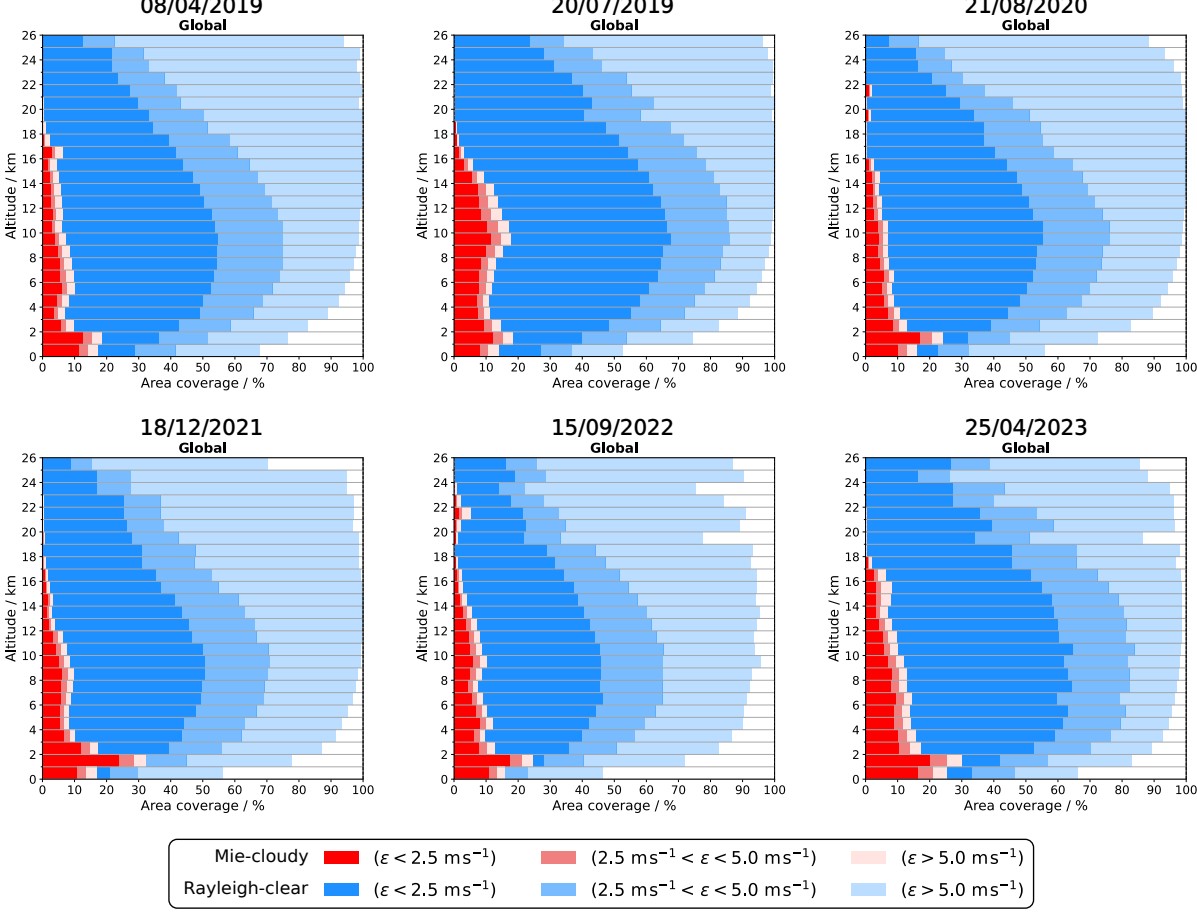

**Figure 5.** Aeolus Mie-cloudy (red) and Rayleigh-clear (blue) global wind data coverage for selected days throughout the mission period. Mie-cloudy coverage is overlaid on top of the Rayleigh-clear coverage. The colour shading indicates the proportion of data within specific intervals of the (O-B) wind speed difference $\varepsilon$ relative to the ECMWF model background, assuming a background error of 2.5 m s$^{-1}$. The Rayleigh-clear wind error is normalised to a vertical bin thickness of 1 km.





Among the different days, the highest coverage occurred in July 2019, when high-quality Rayleigh winds ($\varepsilon < 2.5 \mathrm{\,m\,s^{-1}}$) covered 30–65 % of the measurement range from the surface up to 26 km altitude. When considering all Rayleigh-clear winds, regardless of quality, near-complete coverage was achieved above 10 km, gradually decreasing to about 50 % near the surface.

Interestingly, despite the significant signal loss during the FM-B phase (2019–2022), total Rayleigh wind coverage remained
relatively robust. By September 2022, the reduced SNR and the resulting increase in wind random error had lowered the median coverage across all altitudes from 96 % in July 2019 to 90 %–a relatively modest decrease, considering the 70 % decline in Rayleigh backscatter signal. However, the availability of high-quality Rayleigh winds declined more substantially. At the 10 km peak altitude, high-quality winds were available over only 40 % of the globe, compared to 65 % in July 2019. This reduction corresponds to a rise in the global random wind error, which increased from approximately 5 to 8 $\mathrm{\,m\,s^{-1}}$(see
Fig. 2).

The coverage of Mie-cloudy wind data exhibits greater variability, both with altitude and over the course of the mission, compared to Rayleigh-clear winds. In all datasets, the highest coverage is observed around 2 km altitude, which can be attributed to the presence of low-level liquid clouds and aerosols within the PBL. At this altitude, Mie-cloudy coverage can reach up to 30 %. However, overall Mie wind coverage remains significantly lower than that of Rayleigh winds, with median values
across all altitudes between 7 % and 11 %. Notably, most of the available Mie-cloudy winds are classified as high-quality (see dark-red bars in Fig. 5). This aligns well with the observed stable Mie wind random error, which remained around 3 $\mathrm{\,m\,s^{-1}}$ throughout the mission (see Fig. 2), which is close to the mission requirement of 2.5 $\mathrm{\,m\,s^{-1}}$ for the free troposphere (2–16 km), but above the mission requirement of 1 $\mathrm{\,m\,s^{-1}}$ for the PBL (0–2 km) (European Space Agency (ESA), 2016).

To facilitate comparison among datasets, the altitude-dependent coverage of high-quality winds is shown in consolidated
plots in Fig 6. For Mie-cloudy winds, coverage increases throughout the troposphere and peaks between 1 and 2 km. The drop in coverage at altitudes below 2 km can be explained by attenuation from clouds as well as ground returns in this region which are flagged as invalid wind data in the L2B product, but are considered in the reference area for the calculation of the wind data coverage. Within the troposphere; however, coverage shows significant variability among the six selected days influenced by the presence of clouds, without any consistent trend over the mission duration. One exception is the July 2019 dataset, which
exhibits enhanced coverage between 8 and 16 km. A hemispheric split (dashed and dash-dotted light-blue lines in Fig. 6 (a)) reveals that this enhancement is limited to the Northern Hemisphere (NH) and can be attributed to the presence of aerosols from widespread wildfires, as will be elaborated later in the text. For the other datasets, dividing coverage between NH and the Southern Hemisphere (SH) does not reveal significant differences (e.g., see dark-red lines for the 2023 dataset).

Another noteworthy feature is the increased Mie-cloudy wind coverage above 20 km altitude observed in the 2022 dataset.
This enhancement originates from stratospheric aerosols, which provided sufficient backscatter for Mie wind retrievals at these higher altitudes. Specifically, the Hunga Tonga–Hunga Ha'apai eruption on 15 January 2022 led to a massive injection of volcanic material into the stratosphere, where it persisted for several months (Legras et al., 2022).

Panel (b) of Fig. 6 displays the altitude-dependent coverage of high-quality Rayleigh-clear winds, highlighting the decline from July 2019 (light-blue) to September 2022 (orange). Following the switch back to the FM-A laser in late 2022, which
restored transmission along the instrument emission path and increased atmospheric backscatter, the Rayleigh wind coverage





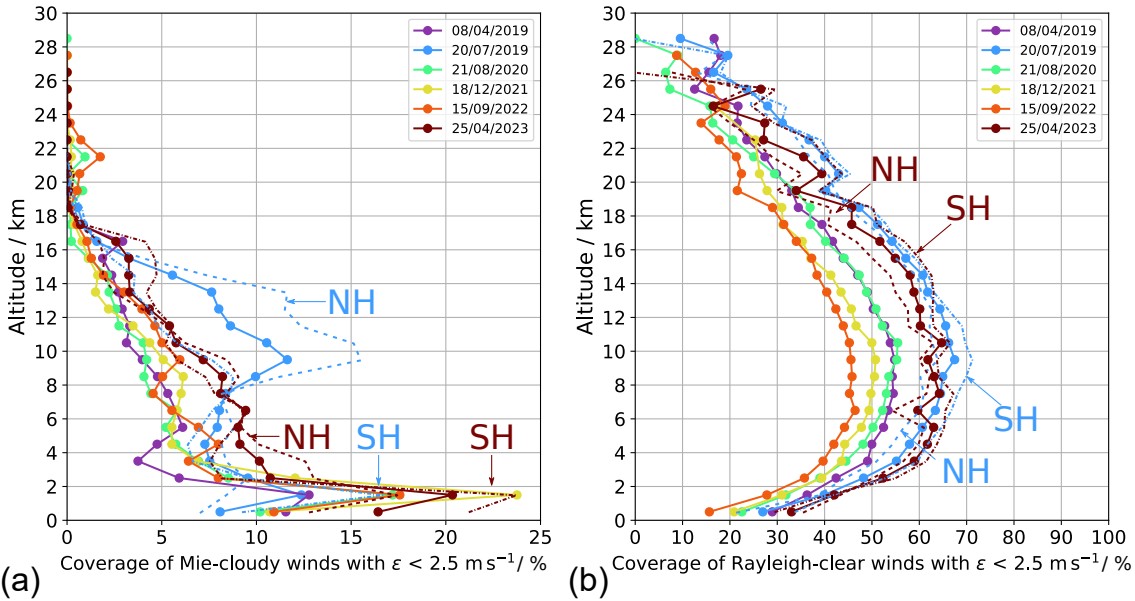

**Figure 6.** Altitude dependence of (a) L2B Mie-cloudy and (b) L2B Rayleigh-clear wind data coverage with (O-B) wind speed difference $\varepsilon < 2.5\,\mathrm{m\,s^{-1}}$ for six selected days throughout the mission period. The Rayleigh-clear wind error is normalised to a vertical bin thickness of 1 km. For 20 July 2019 and 25 April 2023, the data are additionally shown separately for the NH (dashed lines) and SH (dash-dotted lines).

nearly returned to levels seen during the peak performance period in 2019. In contrast to Mie-cloudy winds, no significant hemispheric differences in Rayleigh wind coverage were observed.

### 3.1.2 Coverage in different latitude bands

The coverage of Rayleigh-clear and Mie-cloudy wind observations from the best-case scenario in July 2019 is analysed across three latitude bands and depicted in Fig. 7: (a) the polar region (latitudes above $60°$), (b) the tropics (latitudes below $20°$), and (c) the storm track region (latitudes between $40°$ and $60°$). The data show that the coverage of high-quality Rayleigh-clear winds is greatest in the tropics, particularly at higher altitudes despite the presence of high-altitude convective clouds, primarily due to reduced solar background radiation compared to the poles. In contrast, the presence of strong solar background near the North Pole during the boreal summer introduced significant noise into the Rayleigh signal, thereby degrading the precision and coverage of high-quality Rayleigh wind measurements at higher altitudes in this region.

The Mie-cloudy wind coverage profiles reveal that maximum coverage in the upper troposphere, associated with cirrus and convective clouds, occurs at higher altitudes in the tropics compared to the polar regions, consistent with expectations. Moreover, due to weaker convective activity and drier atmospheric conditions, low-level cloud occurrence is reduced in the polar regions compared to the tropics, where extensive marine stratocumulus decks provide a major source of low-level Mie winds. As anticipated, the storm track region exhibits intermediate coverage characteristics between the poles and tropics. The





wind data coverage in these latitude bands is particularly important, as Aeolus measured the strongest winds in this region, making it highly relevant for NWP data assimilation. Previous studies have demonstrated that, in individual cases, large and spatially coherent wind errors exceeding $10 \, \mathrm{m \, s^{-1}}$ can be found in the analyses and short-range forecasts of both ECMWF and the UK Met Office, especially above the tropopause in upper-level ridges of the North Atlantic jet stream (Schäfler et al., 405   2020). These findings underscore the importance of high-quality wind observations in the storm track region.

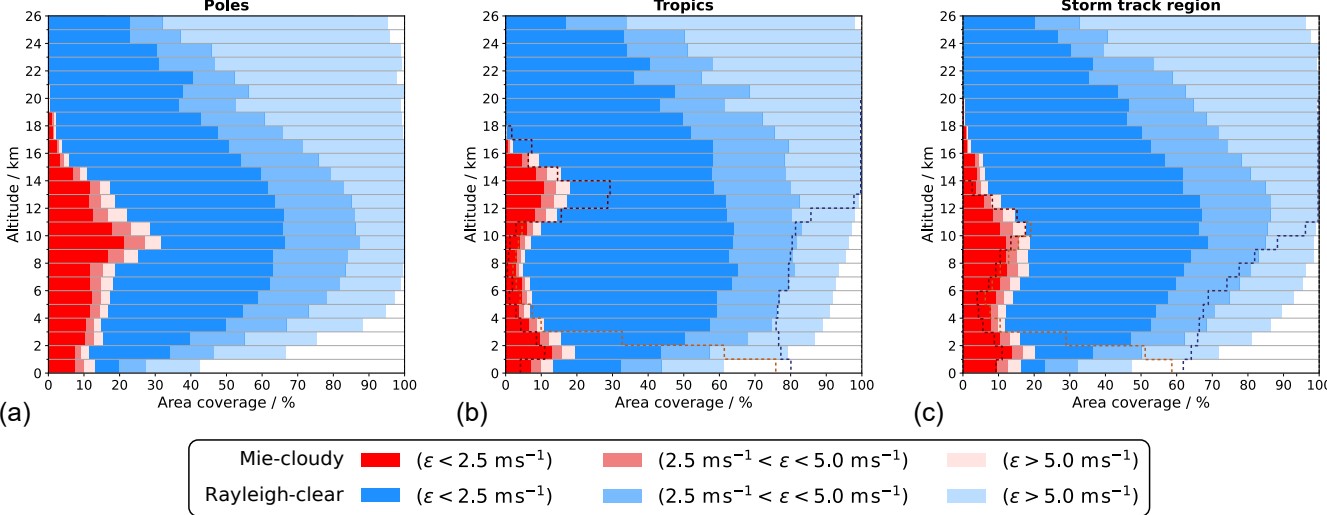

**Figure 7.** Aeolus Mie-cloudy (red) and Rayleigh-clear (blue) global wind data coverage on 20 July 2019 in different geographical regions: (a) poles (latitude $> 60°$); (b) tropics (latitude $< 20°$); storm track region ($40° <$ latitude $< 60°$). The colour shading indicates the proportion of data within specific intervals of the absolute value of (O-B) wind speed difference $\varepsilon$. The Rayleigh-clear wind error is normalised to a vertical bin thickness of 1 km. The dashed lines in panels (b) and (c) represent pre-launch LIPAS simulations of the coverage from clouds (red), aerosols (brown) and molecules (blue), adapted from Marseille et al. (2001).

The observed wind data coverage can be compared to pre-launch simulations generated using the LIPAS (Lidar Performance Analysis Simulator) software (Marseille and Stoffelen, 2003). Given vertical profiles of atmospheric parameters (temperature, pressure, wind speed and direction, cloud fraction, and aerosol concentration) as input, LIPAS produces simulated Aeolus HLOS winds along with corresponding estimates of measurement accuracy. The simulated coverage from a 30-day period (5 410   February to 7 March 1993) (Marseille et al., 2001) is shown in panels (b) and (c) of Fig. 7 as dashed lines. Comparison with in-orbit observations reveals generally good agreement for cloud returns, i.e., the Mie winds, with observed coverage values around 10 %. However, the simulations predicted a higher Mie wind data coverage in the upper tropical troposphere due to convective clouds, reaching nearly 30 %, compared to the observed 20 %. In contrast, Rayleigh wind data from molecular backscatter were underestimated in the simulations below 12 km altitude, where signal attenuation due to thick clouds was 415   expected to be more significant.





The LIPAS simulations also include predicted wind coverage from aerosol backscatter, represented by brown dashed lines. The main aerosol contribution occurs within the PBL below 3 km, with coverage reaching up to 76 % near the tropical surface and about 60 % in the storm track region. In contrast, the Aeolus L2B product shows that the PBL is predominantly covered by Rayleigh-clear winds, while Mie-cloudy winds contribute only up to 20 %. This is because aerosol backscatter is relatively

weak compared to cloud backscatter and, in most cases, too noisy for the horizontal bin lengths of 10–20 km, preventing the retrieval of valid Mie-cloudy winds. While longer accumulation lengths (e.g., 86 km) dedicated to "Mie-aerosol" retrievals would in principle be feasible, the L2B processor was optimized for maximum horizontal resolution without degrading data quality, thereby retrieving mainly winds from cloud backscatter. Simultaneous retrieval of both Mie-cloudy and Mie-aerosol winds would require different accumulation lengths within a single processing run and thus an adaptation of the L2B processor,

which could be considered for Aeolus-2.

Beyond the lack of Mie winds in aerosol-dominated regions, the inclusion of aerosol backscatter in the Rayleigh signal (Mie contamination) can introduce wind biases. Combined with increased signal attenuation, this results in elevated Rayleigh-clear random error at lower altitudes (Dabas et al., 2008). In these regions, however, high-quality Rayleigh-cloudy winds can be retrieved, where Mie contamination is corrected in the L2B processor using scattering ratio estimates and long-term NWP

model data as a reference (Marseille et al., 2023).

For Aeolus-2, a more flexible discrimination between clear-air, aerosol-laden, and cloudy regions is recommended, with adaptive horizontal (and possibly vertical) binning lengths tailored to the SR regime. Such an approach would enhance the effective coverage of Mie winds, which generally provide higher precision than Rayleigh winds, particularly in the PBL where signal attenuation and Mie contamination reduce Rayleigh wind quality.

### 3.1.3 Range bin settings

When comparing Aeolus wind data coverage across different mission phases and geographic regions, it is important to account for the configuration of the 24 atmospheric range bins used in the Rayleigh and Mie channels. The thickness of each range bin were configured independently for both channels, typically set to 250 m, 500 m, 1000 m, or 2000 m, and was adjustable along the orbit (see also Fig. 4). These range bin settings (RBS) affect not only the vertical resolution but also the Rayleigh-clear

random error, which scales inversely with the SNR and, consequently, with bin thickness. To ensure consistent treatment of the wind data in the coverage analysis, the Rayleigh-clear random error was normalised to a bin thickness of 1 km when classifying wind data quality in the individual 1 km altitude bins. When this normalisation is applied, Rayleigh wind data coverage appears relatively homogeneous across the globe, particularly at altitudes above 5 km, and largely reflects the altitude dependence of the SNR and the cloud coverage. For this reason, the following discussions focus on Mie-cloudy wind data coverage.

The RBS of the Mie channel for two selected orbits, on 20 July 2019 and 15 September 2022, are shown in Fig. 8. The horizontal lines represent the top altitudes of the 24 atmospheric range bins, plotted as a function of latitude. The range bin altitudes were adapted to the underlying ground elevation by using a Digital Elevation Model in the commanding on-board the satellite to ensure maximum vertical coverage of the atmosphere along the orbit. In July 2019, the bin thickness for Mie-cloudy winds was kept constant along the orbit, with 1 km-thick bins spanning from 2 km to approximately 20 km altitude, and





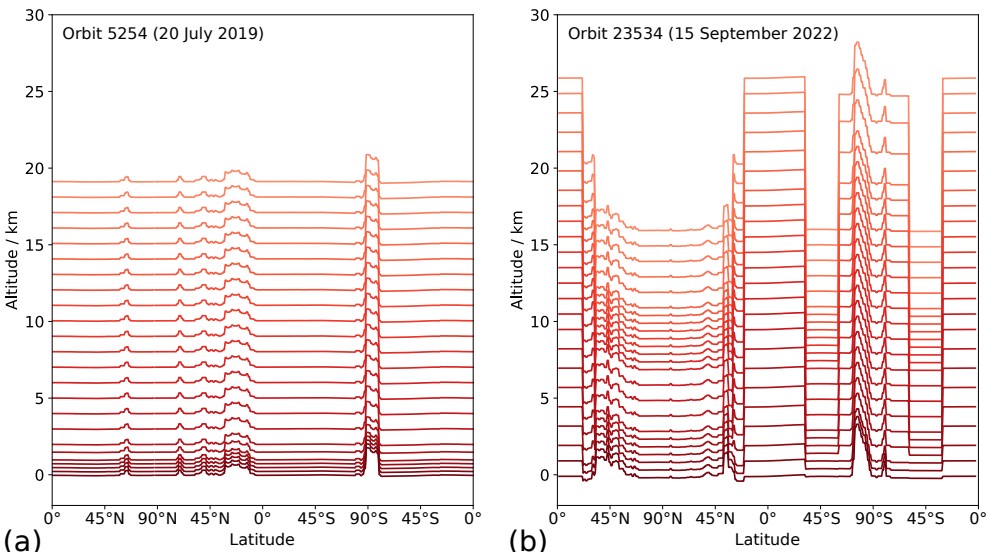

**Figure 8.** Top altitudes of the 24 atmospheric range bins of the Mie channel as a function of latitude along a selected orbit on (a) 20 July 2019 and (b) on 15 September 2022.

narrower bins of 500 m and 250 m near the surface. In contrast, the RBS in September 2022 varied significantly along the orbit: shorter bins were used near the North Pole, where Mie winds were retrieved only up to 17 km altitude, while thicker bins were applied in the tropics, extending up to 26 km to capture the Hunga-Tonga plume. Additionally, in the latitude band between 60°S and 90°S (around the South Pole), thicker range bins were selected to enable detection of polar stratospheric clouds (PSCs) during austral winter and early spring. A similar extension of the range bins to higher altitudes over the South Pole was also applied in 2019, but only from 12 August onward.

### 3.1.4 Mie-cloudy wind data coverage around the globe

To gain a more detailed view of Mie-cloudy wind data coverage around the globe, the dataset was analysed in 5° x 5° latitude/longitude boxes and within five distinct altitude layers: $(2\pm1)$ km, $(5\pm1)$ km, $(10\pm1)$ km, $(15\pm1)$ km, and $(22\pm1)$ km. For robust statistics, extended datasets covering the four-month periods from 1 June to 30 September in both 2019 and 2022 were evaluated. It should be noted that no wind data were available between 16 and 28 June 2019 due to the switch from laser transmitter FM-A to FM-B. The resulting maps of Mie-cloudy wind data coverage, filtered for wind errors $\varepsilon < 2.5 \text{ m s}^{-1}$ (high-quality winds only), are shown in Fig. 9 for both years.

The maps highlight significant variations in Mie data coverage, which are primarily governed by cloud occurrence across different latitudes and altitudes (Feofilov et al., 2022). At low altitudes around 2 km, cloud cover is extensive over oceans, particularly associated with marine stratocumulus decks, whereas continental regions and deserts (e.g., the Sahara and Australian



interior) show markedly lower cloud cover, resulting in reduced Mie wind data coverage. On a global average, high-quality Mie-cloudy winds covered about 15 % of the atmosphere in this altitude range.

At 5 km altitude, Mie-cloudy wind data coverage drops to 8–10 %, concentrated in the tropics where mid-level clouds typically originate from decaying convective systems and elevated moisture layers. A notable contrast between 2019 and 2022 emerges at 10 km altitude. In 2019, enhanced Mie data coverage is apparent in the Arctic, likely due to elevated aerosol loads from widespread wildfires in Alberta, Canada (Shang et al., 2024) and Siberia (Voronova et al., 2020). These aerosols produced narrowband backscatter signals of sufficient strength to enable the retrieval of a substantial number of high-quality Mie winds around the Arctic. In contrast, Mie wind data coverage in the SH at this altitude is primarily driven by upper-level cloud cover, particularly in association with jet streams and frontal systems. This hemispheric disparity is also reflected in Fig. 6. Due to the additional Mie winds around the Arctic, global Mie wind coverage at 10 km altitude reached 11.1 %, nearly double the 5.9 % observed in 2022.

At 15 km altitude, global coverage patterns are similar in both years, with maxima in the tropical regions, particularly over equatorial landmasses within the Intertropical Convergence Zone (ITCZ) and monsoon areas, consistent with the vertical extent of deep convection. At altitudes above 20 km, the data coverage is also influenced by the range bin settings (RBS). In 2019, these were configured to sample this altitude band only near the South Pole, resulting in wind data primarily between the southern tip of South America and Antarctica. This region coincided with the presence of PSCs during austral winter. The relatively high coverage values of up to 30 % can be explained by the strong backscatter of PSCs at 355 nm which is twice as large as that of 532 nm (Jumelet et al., 2009). This enhanced backscatter is largely attributed to the fact that about 50 % of PSCs contain droplets composed of super-cooled ternary solution, which are highly efficient scatterers at ultraviolet wavelengths.

In contrast to 2019, the 2022 RBS enabled Mie wind measurements not only over Antarctica but also across the latitude band from 30°S to 20°N (see Fig. 8). This allowed for the detection of long-lived stratospheric sulfate aerosols from the Hunga Tonga volcanic eruption in January 2022. Initially confined between 30°S and 10°N until May, the ash plume gradually spread southward and descended from ≈23 km to below 20 km altitude (Legras et al., 2022; Duchamp et al., 2023). The adaptation of the Mie RBS in early 2022 was specifically intended to track the temporal evolution of this plume (Trapon and Baars, 2025).





**Figure 9.** Maps of the Mie-cloudy wind data coverage with errors $\varepsilon < 2.5 \ \mathrm{m\,s^{-1}}$ at different altitudes for the periods June–September 2019 (left) and 2022 (right).



In addition to enabling the investigation of the plume's dynamics, the RBS adjustment also yielded high-quality Mie wind
retrievals in the tropical stratosphere, capturing up to 10 % coverage in the altitude range around 22 km within this region.

The Mie wind data coverage for the four-month periods in 2019 and 2022 is presented as a function of latitude and altitude in
Fig. 10. The figure illustrates the zonal mean distribution of high-quality Mie winds, showing maxima in the lower troposphere,
particularly in the SH, and in the upper tropical troposphere, consistent with the cloud cover patterns discussed above. These
features align well with the latitudinal-altitudinal cloud cover distributions reported by Feofilov et al. (2022), which were
derived from both Aeolus and the Cloud-Aerosol Lidar with Orthogonal Polarisation (CALIOP) onboard the Cloud-Aerosol
Lidar and Infrared Pathfinder Satellite Observation (CALIPSO) during the period from 28 June to 31 December 2019. In that
study, a SR threshold of 5 was used at the CALIOP wavelength of 532 nm (after conversion of Aeolus SR to 532 nm) for cloud
detection, meaning that Mie winds associated with aerosols are not represented in their cloud cover plots.

By contrast, the coverage maps in Fig. 10 clearly reflect the influence of aerosols, revealing signatures from wildfire smoke
in the Arctic during 2019 and the Hunga Tonga volcanic plume in the stratosphere over the tropical and subtropical SH in 2022.
The absolute difference between the two years (panel (c)), highlights the gain and loss of high-quality Mie winds resulting from
these two specific aerosol-related events. Apart from these features, the differences between 2019 and 2022 remain relatively
small (5–10 %), indicating that the signal loss of ≈70 % between the periods did not substantially impact the coverage of high-
quality Mie winds. A minor loss of a few percent is, however, noticeable in the mid- to upper troposphere of the SH extratropics,
most likely linked to fewer Mie winds from ice clouds, which produce weak backscatter signals and were therefore detected
less frequently under the reduced signal conditions. Larger differences in the tropics at around 4 km and near the surface can
be attributed to RBS changes. In particular, the use of thicker bins (1 km instead of 500 m or 250 m) in the 30°S–20°N latitude
band in the lower troposphere led to increased Mie wind coverage, albeit at the expense of vertical resolution.

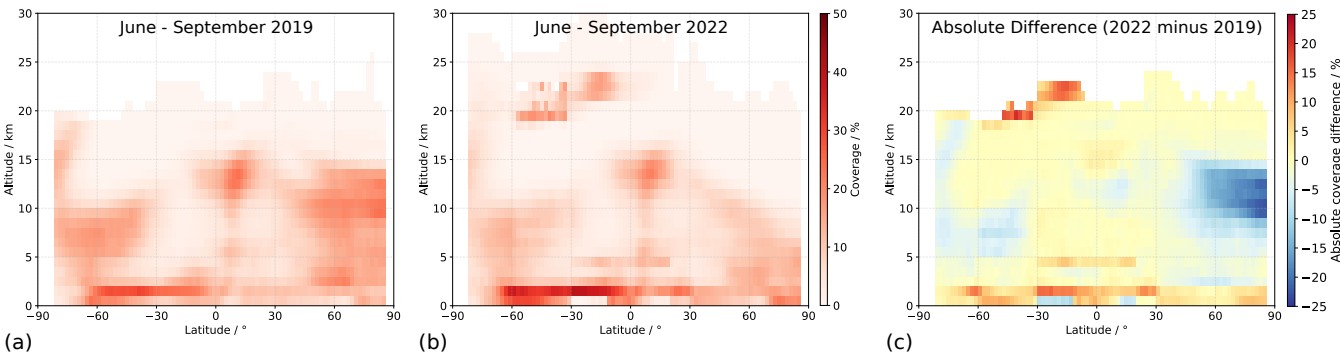

**Figure 10.** Coverage of Mie-cloudy wind results with errors $\varepsilon < 2.5\ \mathrm{m\,s^{-1}}$, shown as a function of latitude and altitude for the periods
June–September 2019 (a) and 2022 (b). Panel (c) shows the absolute difference in percentage coverage between the two years in those
latitude/altitude bins that are available in the datasets of both years.




## 3.2 Signal-to-noise ratio and relation to the Aeolus wind random error

After examining Aeolus wind data coverage across different error regimes, this section shifts focus to the wind error itself and its relationship with the SNR. As described in Sect 2.2.3, correlating the L1B SNR values–defined differently for the Rayleigh and Mie channels–with the L2B wind results requires harmonising the data grids of the two products. To this end, the SNR values from all measurements contributing to a wind result are quadratically averaged (Eq. (1)) to derive the RMS SNR, which can then be correlated with the wind retrieval. The definitions of the Mie and Rayleigh SNRs are provided in Appendix B, which also presents histograms of these parameters for selected mission days.

The Rayleigh total SNR (Eq. (B6)) and the Mie refined SNR (Eq. (B7)), given on measurement-scale (N = 30), are depicted for the previously discussed orbit (orbit 5254 from 20 July 2019) in Fig. 11. In addition to computing the RMS from the SNR values of Rayleigh channels A and B, the total SNR was normalised to a vertical bin thickness of 1 km to account for the different integration lengths associated with the RBS.

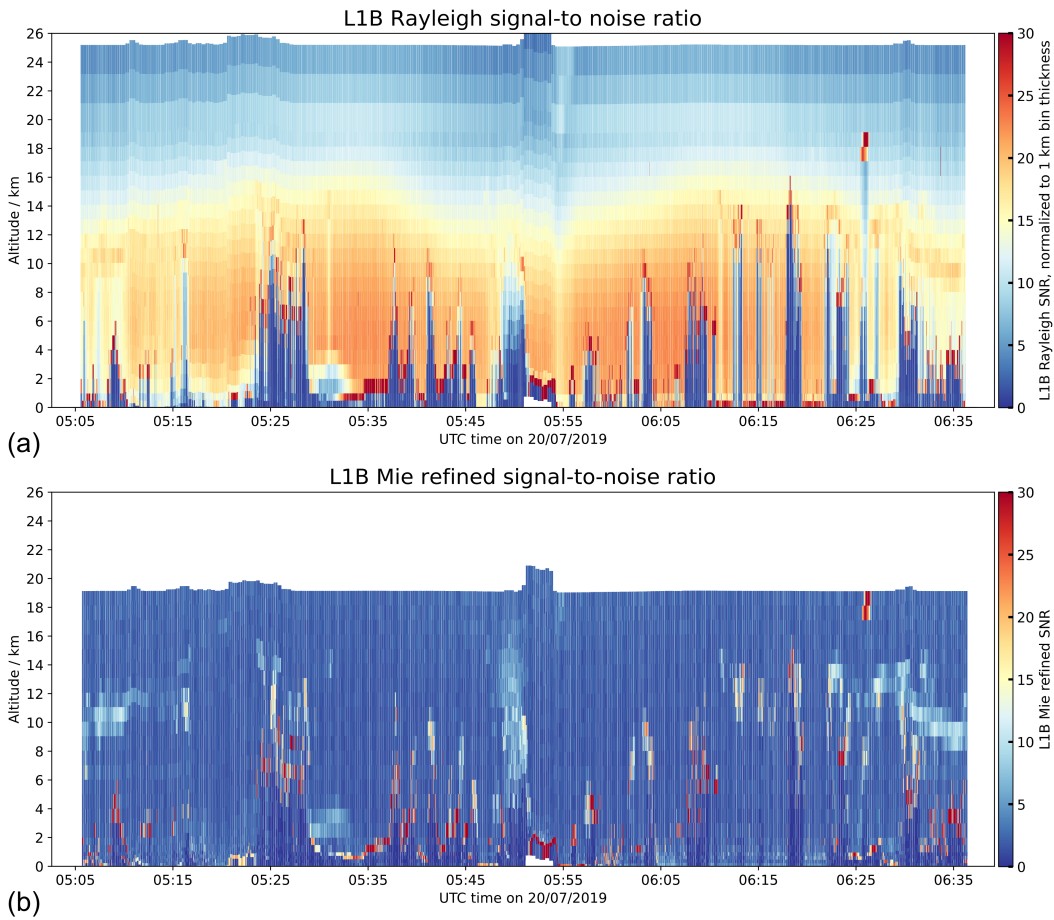

**Figure 11.** (a) Aeolus L1B Rayleigh total SNR and (b) L1B Mie refined SNR (b) on measurement level, acquired on 20 July 2019 between 05:05 UTC and 06:36 UTC (orbit number 5254). The Rayleigh SNR was normalised to a vertical bin thickness of 1 km.



The Rayleigh SNR distribution along the orbit and across altitudes reflects the increasing molecular density from the stratosphere toward the surface. Lower SNR values are observed around the poles (beginning, middle, and end of the orbit) compared to tropical regions at the same altitude. SNR is notably reduced below clouds, often falling below 5, while reaching values up to 25 in the lower troposphere under clear-sky conditions. The Mie refined SNR, shown in panel (b), remains below 5 for most measurement bins and typically increases to 10–20 in the presence of clouds. In cases of strong backscatter from optically thick clouds, the refined SNR exceeds 30, and in rare instances, can surpass 50. Because the Rayleigh and Mie channels rely on fundamentally different measurement techniques, their respective SNR values are not directly comparable.

Figure 12 shows the altitude dependence of the Mie and Rayleigh SNR per wind result. The Mie SNR curves demonstrate a more or less stable pattern over time, increasing from values around 5 above 20 km altitude to more than 20 near the surface. A consistent maximum appears around 2 km, coinciding with the highest coverage of high-quality Mie-cloudy winds, as seen in Fig. 5. The variability in Mie SNR with altitude across the different datasets primarily reflects daily changes in cloud and aerosol cover. Conversely, the altitude dependence of the Rayleigh SNR is mainly governed by the atmospheric molecular density profile and signal attenuation, resulting in a similar shape across all six datasets. The amplitude of the curves correlates with both the atmospheric backscatter signal (see Fig. 2) and the prevailing solar background conditions–being strongest during NH summer. As a result, April 2023 shows similar, or even higher, SNR values compared to July 2019, despite the weaker backscatter signal later in the mission. This improvement is also partly due to the reduction in readout noise following the switch to N = 5 in 2022. Notably, the shape of the Rayleigh SNR profile closely resembles that of the data coverage for high-quality Rayleigh winds (Fig. 6), highlighting that the Rayleigh wind error is predominantly driven by the SNR. This aspect will be examined in more detail in the following sections.

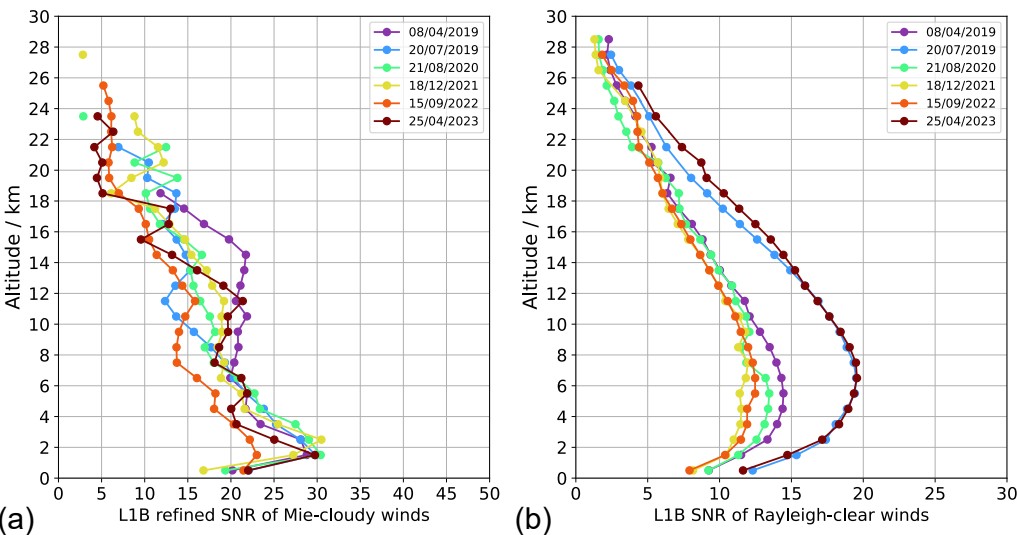

**Figure 12.** Altitude dependence of (a) L1B refined SNR of Mie-cloudy wind results and (b) L1B SNR of Rayleigh-clear wind results for six selected days throughout the mission period. The Rayleigh SNR is normalised to a vertical bin thickness of 1 km and 18 laser pulses per measurement.





### 3.2.1 Wind error estimates

In the L2B product, each Rayleigh-clear and Mie-cloudy wind result is assigned an error estimate (EE). The EE represents the standard error ($1\sigma$) of the HLOS wind speed based on error propagation and can be interpreted as a confidence interval. It is calculated differently for the Rayleigh and Mie channels, reflecting the distinct measurement techniques used to derive the wind from the Doppler frequency shift, as described in Sect. 2.1. A comprehensive explanation of how the EE is computed for both wind types is provided in the Aeolus L2B Algorithm Theoretical Basis Document (Rennie et al., 2020), with a more concise overview available in Lux et al. (2022).

The calculation of the Rayleigh EE is based on the SNR, which incorporates Poisson noise contributions from the atmospheric signals on channels A and B, as well as from the solar background. Since Baseline 16, additional noise from the DCO and detector readout have also been included, as detailed in Appendix B1. The EE then scales inversely with the SNR, as defined in Eqs. (B1-B6):

$$\text{EE}_{\text{Rayleigh-clear}} \propto \frac{1}{\text{SNR}_{\text{Ray,total}}}. \tag{3}$$

For the Mie channel, where the Doppler shift is derived from a Lorentzian fit to the interference fringe imaged onto the detector, determination of the EE is less straightforward than for the Rayleigh channel. Instead of relying on the Mie refined SNR, the EE is computed from the solution of the error covariance matrix of the fit algorithm. This covariance matrix is derived from the partial derivatives of the Lorentzian or Voigt line shape with respect to four fit parameters: peak position, height, width, and offset. As a result, the Mie EE is not directly linked to atmospheric signal levels but depends on the shape of the signal across the Mie detector and how well the fringe matches the fitted line shape.

Histograms of the EE per Mie-cloudy and Rayleigh-clear wind result for the six datasets are shown in Fig. 13. As with the Rayleigh SNR, the Rayleigh EE (panel (b)) was normalised to a bin thickness of 1 km. Its distribution features a steep peak and a long tail. For high-SNR datasets (July 2019 and April 2023), the Rayleigh EE peaks around $3 \, \text{m s}^{-1}$ and extends up to $10 \, \text{m s}^{-1}$. As expected from Eq. (3), the EE shifts to larger values for low-SNR datasets, ranging between 5 and $12 \, \text{m s}^{-1}$. Notably, a small fraction (<1 %) of Rayleigh-clear winds show unrealistically high EE values, up to several tens or even $100 \, \text{m s}^{-1}$ (not shown in the plot), which are addressed later in the text.

In contrast, Mie-cloudy EE distributions (panel (a)) are narrower, typically extending only to about $10 \, \text{m s}^{-1}$. Excluding the dataset from 15 September 2022, these distributions vary little across the mission, pointing to the stable quality of Mie-cloudy winds despite the significant signal loss between 2019 and 2022. The September 2022 dataset stands out with a $2 \, \text{m s}^{-1}$ shift in Mie EE. This is linked to the P/N setting change to 114/5 in April 2022 (Rennie and Isaksen, 2024), where the fact that Mie-cloudy winds were formed from a single L1B measurement may have caused artifacts in the fit covariance matrix. Interestingly, Mie EE values in 2023 decreased again despite the same P/N settings and wind grouping, suggesting a processing issue still under investigation for resolution in Baseline 17.

The altitude dependence of the EE is shown in Fig. 14. For Mie-cloudy winds, EE increases from $2$–$3 \, \text{m s}^{-1}$ in the troposphere to over $5 \, \text{m s}^{-1}$ in the stratosphere, where lower aerosol and PSC backscatter leads to reduced SNR (Fig. 12) and greater




uncertainty in fringe centroid retrieval. Rayleigh-clear EE curves closely mirror the inverse of their SNR profiles: values range from 4–6 $\mathrm{m\,s^{-1}}$ in the troposphere and exceed 10 $\mathrm{m\,s^{-1}}$ in the stratosphere, where molecular signals are weaker and solar

background diminishes the SNR.

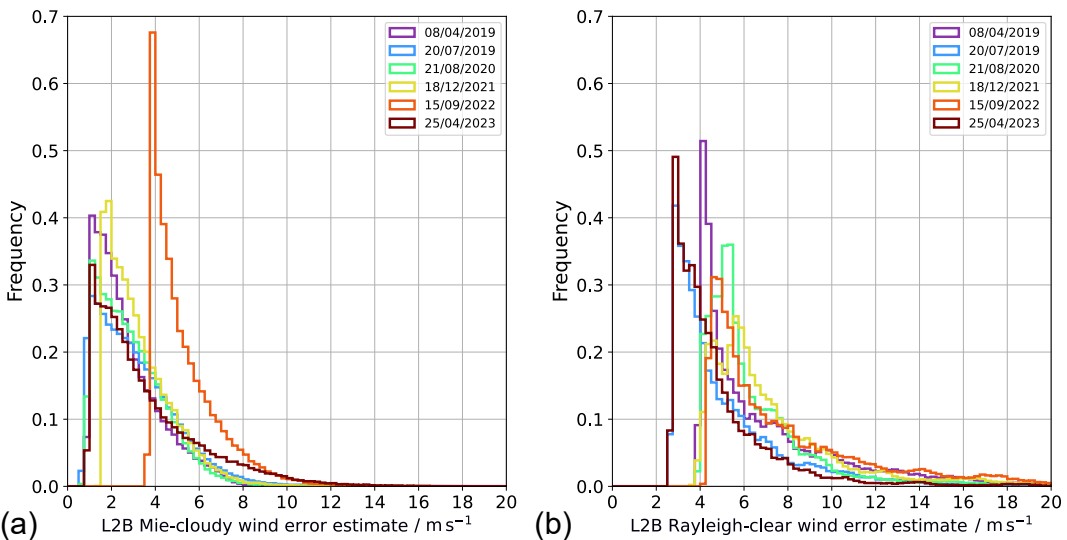

**Figure 13.** Histograms of (a) L2B Mie-cloudy and (b) L2B Rayleigh-clear EE for six selected days throughout the mission period. The Rayleigh-clear EE is normalised to a vertical bin thickness of 1 km.

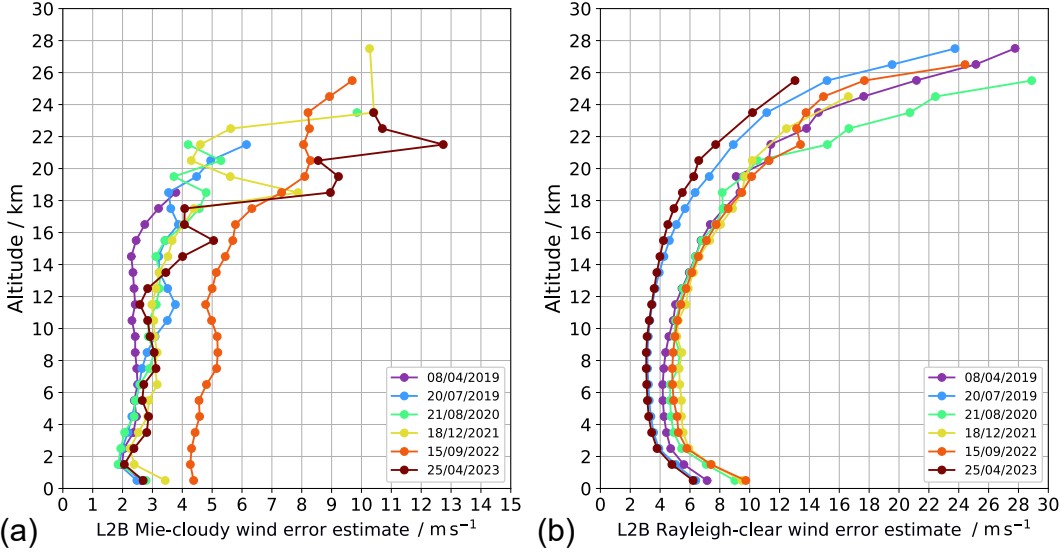

**Figure 14.** Altitude dependence of (a) L2B Mie-cloudy and (b) L2B Rayleigh-clear EE for six selected days throughout the mission period. The Rayleigh-clear EE is normalised to a vertical bin thickness of 1 km.



### 3.2.2 Wind random error with respect to the model

In data assimilation of Aeolus wind products and in many validation studies, the EE is commonly used as a quality control parameter–either to filter out large-error winds or to weight the wind data by quality. Therefore, its representativeness of the actual wind error must be carefully evaluated. To this end, the Aeolus (O–B) wind speed difference between observed winds

and ECMWF model background winds (both included in the L2B product), is compared to the EE. The relationship of both EE and (O–B) wind speed difference with the L1B SNR is shown in Fig. 15 for Mie-cloudy and Rayleigh-clear winds.

Each data point represents the scaled MAD (see Eq. (2)) of the (O–B) wind speed difference, i.e., the wind random error, computed under the assumption of a background error of $2.0 \ \mathrm{m\,s^{-1}}$. Values are aggregated in SNR bins of width 2 for Mie-cloudy winds and 1 for Rayleigh-clear winds. Error bars indicate the range of observation error when background errors of

$1.5 \ \mathrm{m\,s^{-1}}$ and $2.5 \ \mathrm{m\,s^{-1}}$ are assumed. For the EE, the median per SNR bin is shown with dashed lines. Different colours correspond to the selected one-day datasets.

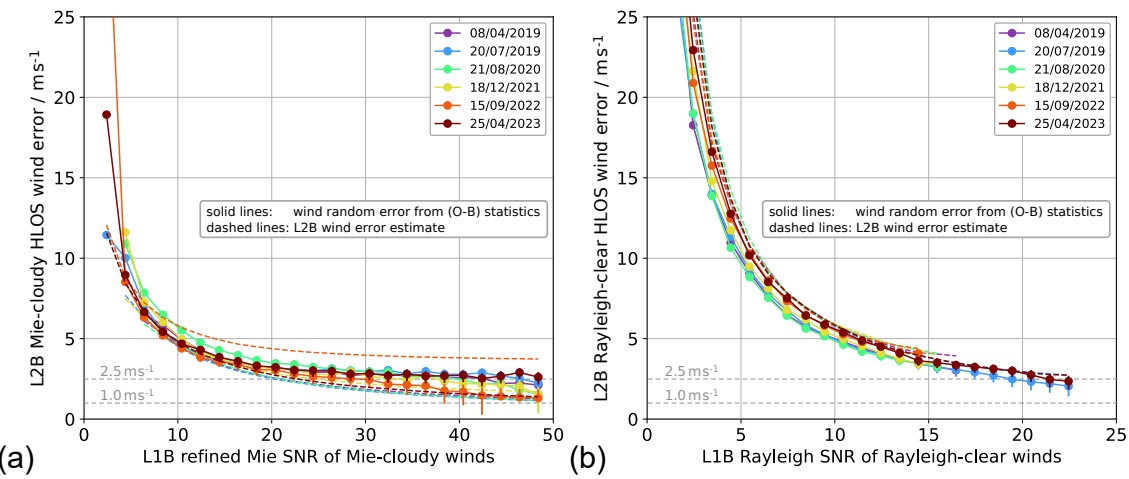

**Figure 15.** L2B wind errors as a function of L1B SNR for the (a) Mie and (b) Rayleigh channels, shown for six selected days throughout the mission period. For Rayleigh, the SNR is normalised to a vertical bin thickness of $1 \ \mathrm{km}$ and 18 laser pulses per measurement. Wind error is expressed as the scaled MAD of the (O–B) wind speed difference, using the ECMWF model background as the reference and assuming a background error of $2.0 \ \mathrm{m\,s^{-1}}$. The error bars indicate the range of observation error when assuming background errors of $1.5 \ \mathrm{m\,s^{-1}}$ and $2.5 \ \mathrm{m\,s^{-1}}$. Dashed lines show the L2B EE, representing the median value per SNR bin (Mie bin width: 2; Rayleigh bin width: 1). The grey horizontal lines indicate the Aeolus mission requirements for the free troposphere (2–16 km) and the PBL (0–2 km), respectively.

Both plots reveal the expected trend of decreasing wind error and EE with increasing SNR, with consistent behaviour across the mission. However, for Mie-cloudy winds, the random error levels off at around $2.5 \ \mathrm{m\,s^{-1}}$ and does not decrease further beyond an SNR of 30 (except for the 2022 dataset). This suggests that the Mie random error is not dominated by

Poisson noise at high SNR but rather by another limiting error source. The fact that the EE continues to decrease, unlike the observed wind error, indicates that this additional noise is not adequately represented in the EE calculation. Currently, orbital





variations in the Fizeau interferometer illumination, driven by thermal fluctuations across the telescope's primary mirror, are assumed to be one contributor to this discrepancy. Indeed, the observed Mie-cloudy random error of 3–4 $\mathrm{m\,s^{-1}}$ (Fig. 2) is notably higher than predicted by wave-optics model simulations, even when accounting for Mie fringe broadening due to imperfect incidence angles and plate defects of the interferometer (Vaughan et al., 2025). Another factor is the underestimation of representativeness error when comparing Mie-cloudy winds to ECMWF model background winds whose effective horizontal resolution is generally coarser (40–80 $\mathrm{km}$) than that of the Mie winds (Rennie et al., 2021).

The Rayleigh-clear EE closely matches the wind random error, especially at higher SNR values, reaching the mission requirement of 2.5 $\mathrm{m\,s^{-1}}$ for SNRs above 20. However, such high SNRs, defined as the total Rayleigh SNR per measurement (18 accumulated pulses), normalised to a 1 $\mathrm{km}$ bin thickness, were only achieved for a portion of wind results during the early FM-B period in summer 2019 and the second FM-A phase in 2023 (see SNR histograms in Fig. B2). For most of the mission, the median Rayleigh SNR was closer to 10, resulting in Rayleigh-clear wind random errors exceeding 5 $\mathrm{m\,s^{-1}}$.

Figure 16 shows the correlation between EE and the wind random error. For Mie-cloudy winds, the EE slightly underestimates the actual error, particularly in the low-error (i.e. high-SNR) regime. An exception is the 2022 dataset, where the change in P/N settings introduced a positive shift in EE, leading to an overestimation of the error, most pronounced at lower values. Although the Mie EE is not derived directly from the refined SNR but from the error covariance matrix, more realistic values–i.e. slightly larger EEs–are expected at low SNR in Baseline 17 once DCO and readout noise are incorporated into the SNR calculation, as is already the case for the Rayleigh channel. In the high-SNR regime, the discrepancy between EE and the wind error derived from (O–B) statistics appears as a flattening of the curve. Extrapolating this curve to EE = 0 $\mathrm{m\,s^{-1}}$ provides an estimate of the representativeness error for Mie-cloudy winds, which is on the order of 2.0–2.5 $\mathrm{m\,s^{-1}}$.

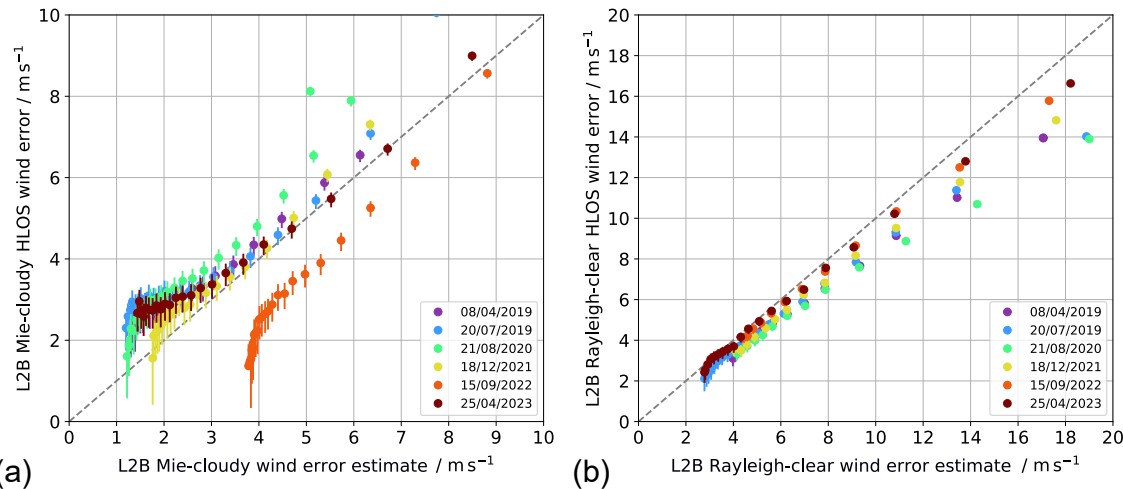

**Figure 16.** L2B wind errors as a function of L2B EE for the (a) Mie and (b) Rayleigh channels, shown for six selected days throughout the mission period. Wind error is expressed as the scaled MAD of the (O–B) wind speed difference, using the ECMWF model background as the reference and assuming a background error of 2.0 $\mathrm{m\,s^{-1}}$. The error bars indicate the range of observation error when assuming background errors of 1.5 $\mathrm{m\,s^{-1}}$ and 2.5 $\mathrm{m\,s^{-1}}$.





For Rayleigh-clear winds, the EE generally represents the wind random error well across most of the range, with significant overestimation occurring only at low SNR. As shown in Fig. 15(b) and consistent with Eq. (3), the EE diverges when SNR drops below 5, yielding unrealistically high values (often several tens of $\mathrm{m\,s^{-1}}$). These low-SNR winds, typically retrieved beneath optically thick clouds (see Fig. 11), are not considered reliable. In Baseline 17, an SNR threshold will therefore be applied to L1B measurements prior to their grouping into Rayleigh-clear winds. This approach simplifies data usage and removes the need for users to apply additional quality filters (e.g. EE thresholds). In the high-SNR (low-error) regime, no flattening of the curve is observed, in contrast to Mie-cloudy winds. This reflects the smaller representativeness error for Rayleigh-clear winds, attributable to longer averaging lengths and more homogeneous backscatter across the range bin. Since the Rayleigh-clear wind random error does not fall well below 3 $\mathrm{m\,s^{-1}}$, where representativeness error would become significant, no flattening occurs.

Finally, Table 2 summarises key statistical parameters describing the Mie and Rayleigh channel performance for the six selected datasets. The Rayleigh median SNR dropped from 13.5 in 2019 to 8.8 in 2022 due to the loss in atmospheric return signal, resulting in a higher wind random error (from 4.5 to 7.0 $\mathrm{m\,s^{-1}}$) and similarly increased EE. While overall Rayleigh-clear wind coverage decreased only slightly (from 96 % to 90 %), the share of high-quality winds ($\varepsilon < 2.5\,\mathrm{m\,s^{-1}}$) declined more noticeably from 74 % to 62 %.

**Table 2.** Statistics of most relevant Aeolus parameters for six selected days throughout the mission period. The Rayleigh SNR, EE and wind random error are normalised to a bin thickness of 1 km. High-quality (HQ) wind data refer to wind results with an (O–B) wind speed difference below 2.5 $\mathrm{m\,s^{-1}}$, assuming a model background error of 2.5 $\mathrm{m\,s^{-1}}$.

| Parameter | 08/04/2019 | 20/07/2019 | 21/08/2020 | 18/12/2021 | 15/09/2022 | 25/04/2023 |
|---|---|---|---|---|---|---|
| Mie wind data coverage | 8.1 % | 11.1 % | 6.7 % | 6.9 % | 7.3 % | 11.2 % |
| HQ Mie wind data coverage | 5.8 % | 9.1 % | 5.4 % | 5.7 % | 5.9 % | 9.0 % |
| Number of Mie winds / day | 77,436 | 139,612 | 83,649 | 58,714 | 61,695 | 93,252 |
| Mie median EE | 2.3 $\mathrm{m\,s^{-1}}$ | 2.8 $\mathrm{m\,s^{-1}}$ | 2.6 $\mathrm{m\,s^{-1}}$ | 2.9 $\mathrm{m\,s^{-1}}$ | 4.8 $\mathrm{m\,s^{-1}}$ | 2.9 $\mathrm{m\,s^{-1}}$ |
| Mie median random error | 3.6 $\mathrm{m\,s^{-1}}$ | 3.9 $\mathrm{m\,s^{-1}}$ | 3.9 $\mathrm{m\,s^{-1}}$ | 3.7 $\mathrm{m\,s^{-1}}$ | 3.9 $\mathrm{m\,s^{-1}}$ | 4.1 $\mathrm{m\,s^{-1}}$ |
| Mie median refined SNR | 22.8 | 18.6 | 20.4 | 20.5 | 16.0 | 19.8 |
| Rayleigh wind data coverage | 95 % | 96 % | 95 % | 96 % | 90 % | 96 % |
| HQ Rayleigh wind data coverage | 68 % | 74 % | 70 % | 68 % | 62 % | 74 % |
| Number of Rayleigh winds / day | 149,687 | 148,666 | 140,912 | 145,028 | 131,992 | 137,011 |
| Rayleigh median EE | 5.9 $\mathrm{m\,s^{-1}}$ | 4.3 $\mathrm{m\,s^{-1}}$ | 5.7 $\mathrm{m\,s^{-1}}$ | 6.5 $\mathrm{m\,s^{-1}}$ | 6.7 $\mathrm{m\,s^{-1}}$ | 4.1 $\mathrm{m\,s^{-1}}$ |
| Rayleigh median random error | 5.7 $\mathrm{m\,s^{-1}}$ | 4.5 $\mathrm{m\,s^{-1}}$ | 5.5 $\mathrm{m\,s^{-1}}$ | 6.4 $\mathrm{m\,s^{-1}}$ | 7.0 $\mathrm{m\,s^{-1}}$ | 4.6 $\mathrm{m\,s^{-1}}$ |
| Rayleigh median SNR | 10.2 | 13.5 | 10.5 | 9.2 | 8.8 | 14.2 |



In contrast, Mie wind coverage was less sensitive to the signal loss and more influenced by cloud and aerosol variability, with notable enhancements following events such as the 2019 wildfires and the 2022 Hunga Tonga eruption. The increase in Mie-cloudy wind coverage in April 2023, despite no major aerosol event, suggests that improved signal transmission also benefits the Mie channel. The combination of longer horizontal accumulation (17 km at N = 5) and stronger backscatter allowed retrievals from weaker aerosol layers that previously fell below the detection threshold. This is especially evident below 10 km,

with a similar increase observed in both hemispheres (Fig. 6).

## 4   Summary and conclusions

The Aeolus mission was a groundbreaking space mission, delivering, for the first time, vertical wind profiles on a global scale. Near-real-time assimilation of wind observations into NWP models at major weather services worldwide led to significant improvements in forecast skill. Beyond operational forecasting, wind and aerosol data were widely used in scientific studies of

atmospheric dynamics, including jet stream variability and structure, the quasi-biennial oscillation, stratosphere–troposphere exchange, volcanic plume tracking, and wildfire smoke monitoring. The accuracy of such analyses critically depends on data quality and availability across different altitudes and geographical regions.

    To assess the observational footprint of Aeolus throughout its mission, the atmospheric area covered by wind observations was computed, accounting for the variable horizontal and vertical extent of wind bins from the Rayleigh and Mie channels, and

analysed in 1 km altitude bins. Wind data were also categorised based on their error relative to the ECMWF model background, incorporating background error estimates. This enabled a detailed evaluation of the altitude-dependent availability of high-quality Mie-cloudy and Rayleigh-clear winds, i.e., those meeting the random error mission requirement, on representative days across the five-year mission. To link wind error from (O-B) departure statistics and the L2B error estimate to the SNR in the L1B product, an averaging algorithm was developed to calculate the RMS-value of the SNR of all L1B measurements contributing

to a single L2B wind. This was particularly important as the number of contributing measurements varied throughout the mission, especially in the Mie channel. Together, these methods provided a comprehensive view of Aeolus data quality and coverage in relation to SNR.

    Wind data coverage and random error varied considerably over the mission, primarily due to a 70 % reduction in detected atmospheric backscatter signal between mid-2019 and late 2022. This degradation mainly affected the Rayleigh channel, where

retrieval precision is dominated by signal Poisson noise. As the median Rayleigh SNR (normalised to 18 accumulated pulses and 1 km vertical bin width) fell from 13.5 to 8.8, the corresponding median random error increased from 4.5 to 7.0 $\mathrm{m\,s^{-1}}$, and the fraction of high-quality winds declined from 74 % to 62 %.

    Conversely, Mie channel SNR remained relatively stable, with a median SNR range of 16 to 23, and random error varied only slightly between 3.6 and 4.1 $\mathrm{m\,s^{-1}}$. The Mie wind error estimate slightly underestimates the actual error, though more realistic

values–particularly at low SNR–are expected in Baseline 17 once DCO and readout noise are included in the SNR calculation. Extrapolation of the EE–random error relationship suggests a representativeness error of about 2.0–2.5 $\mathrm{m\,s^{-1}}$. In addition, an





extra noise source appears to limit the random error in the high-SNR regime, most likely related to orbital variations of the Fizeau illumination that induce signal fluctuations above the Poisson noise limit.

Rayleigh-clear winds provided extensive coverage, exceeding 90 % up to 26 km altitude. Mie-cloudy wind coverage was more variable with altitude, latitude, and season, typically ranging between 5 % and 15 %. It generally followed global cloud coverage but also revealed distinct features in aerosol-rich conditions. One extreme case was the widespread NH wildfires in 2019, which nearly doubled global Mie wind coverage at 10 km altitude from 6 % to 11 % when compared to other years. Similarly, the Hunga Tonga eruption in January 2022 injected long-lived sulfate aerosols into the stratosphere, enabling high-quality wind retrievals well into September across much of the SH.

These findings are highly relevant for Aeolus-2, which entered Phase B2 in 2025 and is currently scheduled for launch in 2034. As part of the EUMETSAT Polar System (EPS), Aeolus-2, also referred to as EPS-Aeolus, is being developed in partnership between ESA and EUMETSAT. Two satellites will be launched in succession, ensuring at least ten years of continuous operation. ESA is responsible for satellite and instrument development, while EUMETSAT handles ground segment development, launch procurement, and satellite operations, including data processing and delivery.

Aeolus-2 will offer enhanced vertical resolution, with 66 atmospheric range bins compared to 24 on Aeolus, and will cover a higher altitude range of up to 40 km. In addition, a higher laser output energy of 150 mJ, combined with reduced signal losses along the emission and reception paths, will result in significantly lower observation errors, especially for Rayleigh-clear winds, due to the improved SNR (Porciani, 2025). Besides broader Rayleigh wind coverage at higher altitudes, the improved instrument performance of Aeolus-2 is expected to enable the retrieval of more Mie winds, particularly from thin clouds and aerosol layers where SNR was insufficient to retrieve valid data during the Aeolus mission.

Prior to the launch of Aeolus-2 in the mid-2030s, the WIVERN (wind velocity radar nephoscope) mission is scheduled for launch in the 2032-33 time frame if selected as ESA's 11th Earth Explorer mission (MPI, 2025). Using a conically scanning Doppler W-band radar, WIVERN will provide direct wind observations within clouds, addressing a major gap in the Global Observing System in cloudy and precipitating areas. Since the two missions will observe winds in complementary regions of the atmosphere, WIVERN and Aeolus-2 data will be synergistic as impressively demonstrated by the cloud-profiling radar and atmospheric lidar on EarthCARE (Wehr et al., 2023; Mason et al., 2024; European Centre for Medium-Range Weather Forecasts, 2025). While WIVERN will sample cloudy regions throughout the troposphere, Aeolus-2 will observe winds in clear-sky and thin-cloud conditions, extending into the stratosphere.

The potential combined impact of both missions on NWP was demonstrated in (Sasso et al., 2025), which showed no saturation effect from their joint assimilation. Instead, their impacts are nearly additive, reducing uncertainty in 3-hour, 6-hour, and 12-hour wind forecasts by up to 6 % when assimilated together. These results are highly promising in the context of the simultaneous operation of these two active wind-measuring satellite missions in the mid-2030s.





## Appendix A: Level-2B processor

The L2B processing of Aeolus L1B wind data is essential for generating L2B data products, specifically retrieving the consoli-
dated horizontal line-of-sight (HLOS) wind component from the Rayleigh and Mie channels. This process applies corrections,
modifications, and averaging techniques to L1B spectrometer signals to produce HLOS wind observations, referred to as wind
results, suitable for assimilation into NWP systems and scientific research. In particular, L2B processing involves the following
steps, using L1B measurement data and calibration products as inputs:

– Measurement bins are classified as *cloudy* or *clear* using the Mie refined SNR from the L1B product to minimise Mie
contamination of the Rayleigh channel for Rayleigh-clear winds and maximise SNR for Mie-cloudy winds.

– Measurements are gathered together into so-called *groups* according to parameter settings to optimise horizontal resolu-
tion and noise characteristics. These *groups* may differ from the standard BRC grouping used in the L1B processing.

– Spectrometer counts from grouped and classified bins are accumulated before wind retrieval to achieve appropriate
resolution and noise levels.

– (H)LOS wind retrievals are performed separately for Mie and Rayleigh channels, yielding four wind types: Rayleigh-
clear, Rayleigh-cloudy, Mie-clear, and Mie-cloudy. Mie-clear is usually not produced, but a few wind results may exist
due to noise in the optical properties information.

– Rayleigh and Mie winds are corrected for orbital variations in the spectrometer incidence angle, which are correlated
with temperature gradients of the primary telescope mirror (M1). This correction uses a multiple-linear regression model
with M1 temperature sensor data from the L1B product and ECMWF model wind data over a 24-hour period to adjust
subsequent wind measurements. (Weiler et al., 2021b; Rennie et al., 2021).

– Rayleigh winds are corrected for temperature and pressure effects using Rayleigh-Brillouin Correction (RBC) (Dabas
et al., 2008), which relies on a priori temperature and pressure information from the AUX_MET product and a pre-
calculated calibration look-up table in the auxiliary RBC input file (AUX_RBC). RBC accounts for the Rayleigh-
Brillouin effect, which deforms the spectral peak shape depending on temperature and pressure (Šavli et al., 2021).

– Rayleigh-cloudy winds are corrected for Mie contamination (cross-talk) using L1B SR estimates (Marseille et al., 2023)
in contrast to Rayleigh-clear winds.

– Error estimates and quality flags are derived to each wind result.

More details of the individual L2B processing steps can be found in Rennie et al. (2020) and The Aeolus Data Innovation
and Science Cluster (DISC) (2024). Notably, the L2B processor sums spectrometer signals from Aeolus measurement bins
(effectively performing horizontal averaging) after (optionally) classifying them as clear or cloudy. Wind retrieval is then
performed on these integrated signals rather than on individual measurements before averaging. Processing algorithm choices,



such as thresholds and switches, are controlled via an auxiliary input parameter settings file (AUX_PAR_2B). A simplified scheme of the Aeolus data products that are investigated in this study, is depicted in Fig. A1.

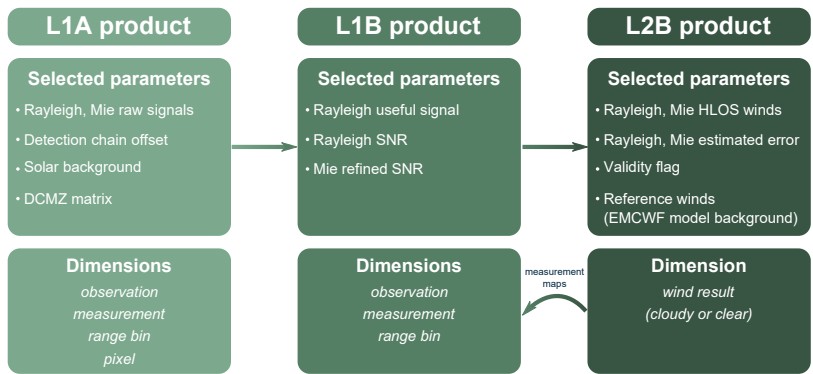

**Figure A1.** Overview of the L1A, L1B, and L2B product parameters analysed in this study, along with their respective dimensions.

## A1 Level-2B grouping algorithm

The L1A and L1B data acquired during wind measurements are structured on a two-dimensional grid. One dimension corresponds to the number of observations, each subdivided into N measurements, while the other dimension represents the 24 atmospheric range bins. Some L1A parameters, such as Mie signal intensities and the dark current values in the memory zone, are also provided for each of the 16 detector pixels along each row of the memory zone, forming a third dimension that is not considered further in this discussion. This two-dimensional L1 data grid can be visualised as a matrix of N vertical profiles, where the horizontal axis represents the measurement number (along-track), and the vertical axis corresponds to the range bin index (altitude). The individual cells within this grid are referred to as measurement bins. The data structure for the Mie and Rayleigh channels is illustrated in the measurement maps in Fig. A2.

In the L2B processor, measurement bins from the Rayleigh and Mie channels are combined by a Grouping Algorithm (GA) to generate the so-called wind results. Unlike L1 data, the L2 output does not consist of wind profiles tied directly to individual observations. Instead, each wind result has a horizontal extent that varies according to the L2B processor settings. This design provides flexibility in defining the horizontal resolution of the wind results, enabling a trade-off between spatial detail and SNR. Longer along-track accumulations may be required to obtain more representative wind estimates for NWP assimilation or research applications, while still maintaining acceptable random error. Consequently, each wind result is assigned its own geolocation information, whereas L1 parameters retain the same geolocation across all range bins of a given measurement.

The GA is applied only in the horizontal direction, leaving the vertical extent of the bins–defined by the range-bin settings of the Rayleigh and Mie channels–unchanged. It is applied independently to the two channels using separate configurations, reflecting their distinct signal properties. In particular, the channels require different horizontal averaging to achieve comparable precision. Mie winds typically require fewer measurement bins than Rayleigh winds due to the stronger atmospheric backscatter from clouds compared to molecular scattering. Throughout the mission, the Rayleigh wind accumulation length



was consistently set to approximately 86 km, meaning that all N measurements from a single L1 observation were grouped into a single L2B wind result. In contrast, early in the mission, the GA for the Mie channel was configured to produce L2B Mie wind results with an along-track extension of approximately 50 km.

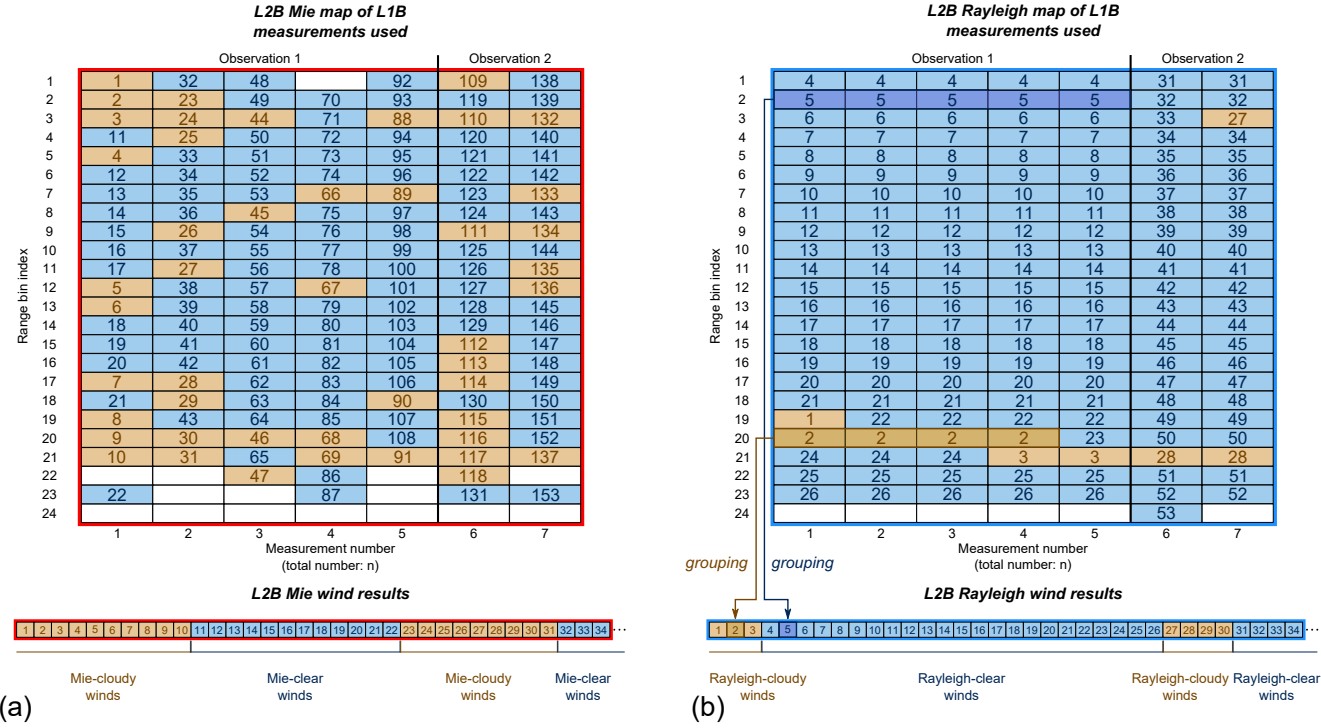

**Figure A2.** Schematic illustrating the grouping algorithm for (a) Mie winds and (b) Rayleigh winds in the Aeolus L2B processor. Individual L1B measurements per range bin and observation are classified as "cloudy" or "clear", gathered in groups according to parameter settings, and then alternately stacked into a 1D array in the L2B product. The L2B wind results are enumerated following the order of measurements within one observation. The scheme shown is based on L1B and L2B data from orbit 21857 (1 June 2022, 05:49 UTC–07:19 UTC) with P/N settings 114/5. Note that for other P/N settings, multiple L1B measurements may contribute to a single L2B Mie wind result. Due to the longer accumulation length of Rayleigh winds compared to Mie winds, multiple L1B measurements contribute to a single wind result, even for P/N settings 114/5. This is exemplary shown for wind result 2 (Rayleigh-cloudy) and wind result 5 (Rayleigh-clear).

On 5 March 2019, this accumulation length was reduced to approximately 11 km, increasing the number of Mie wind results by a factor of two to three while increasing the random error by only about 5 % (Rennie et al., 2021). This minor increase in random error was expected since cloud backscatter was sufficiently strong to avoid signal limitations, in line with pre-launch predictions (Šavli et al., 2019). In reprocessed datasets of the early mission phase before March 2019 (Baselines 14 and 16), the Mie accumulation length is also set to approximately 11 km.

Following the change in P/N settings to 114/5 in April 2022, the minimum accumulation length for Mie wind results increased to ≈17 km (86 km/5), since at least one measurement bin is required to form a L2B wind result. Figure A2 illustrates





the GA for the first seven measurements of orbit 21857, which began on 1 June 2022 at 05:49 UTC. At this stage of the mission, P/N settings of 114/5 were active, implying that each L2B Mie wind result consisted of only one L1B measurement bin.

## A2 Scene classification

Despite the different designs and operating principles of the Mie and Rayleigh channel, the particulate and molecular com-
ponents of the atmospheric backscatter signal cannot be retrieved separately by the two channels. To address this, a scene classification is applied in the L2B processor to distinguish between cloudy and clear-air measurements. Initially, this classification was based on the L1B SR, derived from the Mie channel for each measurement bin. The SR is generally defined as the ratio of the total backscatter coefficient (molecular plus particulate) to the molecular backscatter coefficient: $\mathrm{SR} = \frac{\beta_{mol} + \beta_{par}}{\beta_{mol}}$. Values greater than 1 indicate the presence of particulate backscatter and thus suggest Mie signal contamination in the Rayleigh
measurements. In the L2B processor, the distinction between clear and cloudy scenes was initially made by applying an SR threshold specified in the AUX_PAR_2B configuration file, typically ranging between 1.2 and 1.4. However, despite several refinements to the processor, the SR calculated at the L1B measurement scale was found to be noisy and unreliable for robust classification. As a result, in later processor baselines, the measurement-scale L1B SNR was adopted for classification instead until baseline 16. The definition of this quantity is provided in Appendix B.

The classification of the L1B measurement bins is indicated in Fig. A2 by colour-coding: brown cells for cloudy and blue cells for clear measurements. In rare cases, e.g., when no valid SNR value is available in the L1B product, a bin remains unclassified and is shown as a white cell. The number in each cell refers to the consecutive number of the L2B wind result that the corresponding L1B measurements contribute to. While the Mie-cloudy and Mie-clear wind results are derived from a single L1B measurement for the selected orbit (panel (a)), Rayleigh-clear wind results typically incorporate five measurements
(panel (b)). For Rayleigh-cloudy winds, the accumulation length is more variable to exploit the stronger signal returns from clouds, allowing for higher spatial resolution (Marseille et al., 2023). The cloudy and clear wind results are first numbered and sorted based on the position of the initial contributing L1B measurement within the 2D matrix. They are then alternately stacked, as illustrated in Fig. A2, to form 1D arrays representing the L2B Mie and Rayleigh wind results. The Mie and Rayleigh measurement maps, which link each wind result to its contributing measurement bins, are included in the L2B product and
were used in the data analysis described below.

## Appendix B: Definition of the Rayleigh and Mie SNR

### B1 Definition of the Rayleigh SNR

The L1B product provides SNR values for both the Rayleigh and Mie channels, defined differently due to their distinct optical characteristics. The Rayleigh SNR is computed from the FPI filter A and B signals, corresponding to the sum of signals
on pixels 1–8 for filter A and pixels 9–16 for filter B (see Fig. 1), under the assumption of Poisson noise based on signal levels. However, after launch, it became evident that the Rayleigh SNR is not dominated solely by Poisson noise, as the actual





signal levels were lower than expected by a factor of 2.5 to 3 (Reitebuch et al., 2020). Additional noise sources, such as solar background noise, readout noise, and noise from the detection chain offset (DCO) correction, also contribute significantly. While solar background noise was already accounted for in earlier baselines, readout noise and DCO noise contributions were
incorporated starting from Baseline 16. This led to more realistic, though lower, Rayleigh SNR values. The Rayleigh SNR for channel A of the $i$-th measurement and the $k$-th range bin is defined as:

$$\text{SNR}_{\text{Ray,A}}(i,k) = \frac{S_{\text{A}}(i,k)}{\sqrt{\eta_{\text{signal}} + \eta_{\text{sbkg}} + \eta_{\text{dco}} + \eta_{\text{readout}}}} \quad \text{with} \tag{B1}$$

$$\eta_{\text{signal}} = k_{\text{Ray}} \cdot S_{\text{A}}(i,k) \ , \tag{B2}$$

$$\eta_{\text{sbkg}} = k_{\text{Ray}} \cdot (\frac{t_k}{t_{\text{sbkg}}} + 1) S_{\text{sbkg,A}}(i), \tag{B3}$$

$$\eta_{\text{dco}} = \frac{N_{\text{A}}^2}{N_{\text{dco}}} \cdot \sigma_{\text{dco,Ray}}^2, \tag{B4}$$

$$\eta_{\text{readout}} = N_{\text{A}} \cdot \sigma_{\text{readout,Ray}}^2. \tag{B5}$$

Here, $S_{\text{A}}(i,k)$ is the corrected Rayleigh signal level in channel A of the $i$-th measurement and the $k$-th range bin, after subtracting the DCO, solar background, and dark current in the memory zone (DCMZ). $k_{\text{Ray}} = 0.434 \text{ LSB electrons}^{-1}$ is the radiometric gain from the Rayleigh detection chain. In addition to the Poisson noise of the atmospheric backscatter signal
($\eta_{\text{signal}}$), the Poisson noise $\eta_{\text{sbkg}}$ of the solar background signal $S_{\text{sbkg,A}}(i)$, must be considered. This background signal is scaled by the ratio of the integration time of the range bin $t_k$ to the solar background integration time $t_{\text{sbkg}}$. The DCO noise contribution ($\eta_{\text{dco}}$) scales with the number of pixels contributing to the channel A signal, $N_{\text{A}}$ (typically 8), and the DCO noise level $\sigma_{\text{dco,Ray}} = 2.2 \text{ LSB}$. The number of pixels used for DCO estimation, $N_{\text{dco}}$, also affects this term. Finally, the Rayleigh readout noise ($\sigma_{\text{readout,Ray}} = 2.4 \text{ LSB}$) contributes through $\eta_{\text{readout}}$, which also scales with $N_{\text{A}}$.
In this manner, the Rayleigh SNR is defined separately for each of the two FPI filters. To relate the Rayleigh SNR to L2B parameters, the total SNR is computed as the root-mean-square (RMS) of the Rayleigh SNRs from channels A and B:

$$\text{SNR}_{\text{Ray,total}}(i,k) = \sqrt{\frac{\text{SNR}_{\text{Ray,A}}^2(i,k) + \text{SNR}_{\text{Ray,B}}^2(i,k)}{2}}. \tag{B6}$$

**B2  Definition of the Mie refined SNR**

The Mie SNR computation requires determination of the Rayleigh offset to separate the Mie and Rayleigh signal contributions
across the 16 detector pixels of the Mie detector. Initially, this offset was estimated as the mean of the four pixels with the lowest signal levels. However, this approach proved unreliable in cases of strong Mie backscatter. To improve robustness, the Rayleigh offset was later derived using the so-called Mie Core 2 algorithm, which models the signal as a Lorentzian function superimposed on a constant background. A major enhancement was introduced with Baseline 15 through the Mie Core 3 algorithm, which replaces the Lorentzian with a Voigt function atop a constant offset.



The Mie SNR derived from this fitted function is referred to as the Mie refined SNR, and is defined as:

$$\mathrm{SNR_{Mie,ref}}(i,k) = \frac{1}{\sqrt{k_\mathrm{Mie}}} \cdot \frac{S_\mathrm{Mie}(i,k) - 16 \cdot S_\mathrm{offset}}{\sqrt{S_\mathrm{Mie}(i,k)}}, \tag{B7}$$

where $S_\mathrm{Mie}(i,k)$ is the corrected Mie signal of the $i$-th measurement and $k$-th range bin, after subtraction of the DCO, solar background, and DCMZ. The radiometric gain of the Mie detection chain is $k_\mathrm{Mie} = 0.684\,\mathrm{LSB\,electrons^{-1}}$, and $S_\mathrm{offset}$ is the Rayleigh offset determined from the Mie Core 3 algorithm.

The primary application of the Mie refined SNR is the classification of clear-air and cloudy scenes in the L2B processor, as described in Appendix A2. In Baseline 17, the Mie refined SNR will be extended to include readout and DCO noise contributions, analogous to the Rayleigh SNR formulation. In addition, solar background noise will be taken into account, leading to a definition of the Mie refined SNR that is consistent with the Rayleigh SNR:

$$\mathrm{SNR_{Mie,ref,B17}}(i,k) = \frac{S_\mathrm{Mie}(i,k)}{\sqrt{k_\mathrm{Mie}S_\mathrm{Mie}(i,k) + k_\mathrm{Mie}(\frac{t_k}{t_\mathrm{sbkg}}+1)S_\mathrm{sbkg,Mie}(i) + \frac{N_\mathrm{Mie}^2}{N_\mathrm{dco}}\cdot\sigma_\mathrm{dco,Mie}^2 + N_\mathrm{Mie}\cdot\sigma_\mathrm{readout,Mie}^2}} \ , \tag{B8}$$

analogous to Eq. (B1). Here, $S_\mathrm{sbkg,Mie}(i)$ is the solar background signal on the Mie channel, $N_\mathrm{Mie}$ is the number of contributing pixels (typically 16), and the DCO and readout noise values are $\sigma_\mathrm{dco,Ray} = 3.2\,\mathrm{LSB}$ and $\sigma_\mathrm{readout,Mie} = 3.5\,\mathrm{LSB}$, respectively. When considering the respective radiometric gains $k_\mathrm{Ray}$ and $k_\mathrm{Mie}$, the DCO and readout noise values are consistent across the two channels, corresponding to (4.9±0.2) electrons and (5.3±0.2) electrons, respectively. The latter agrees also, within the uncertainty margin, with the value reported in Lux et al. (2024b).

## B3 Signal distributions and corresponding SNR

Figure B1 shows exemplary Mie and Rayleigh signal levels for selected measurements from orbit 5254, which is discussed in the main text (see, e.g., Fig.11). Curves of the same colour correspond to Level-1B measurements that were grouped together by the L2B processor to produce either a Mie-cloudy or Rayleigh-clear wind result. The Mie signal distribution, shown in panel (a), reveals the characteristic Mie interference fringe, which becomes increasingly pronounced with higher Mie refined SNR, as defined in Eq. (B7). The inset provides the RMS SNR, i.e., the quadratic mean of the refined SNR values across all contributing L1B measurements, as well as the error estimate (EE) of the corresponding L2B Mie-cloudy wind result, which demonstrates the expected anti-correlation with SNR (see Fig. 15).

Panel (b) shows the Rayleigh signal distribution of selected measurements from the same orbit, with the two peaks corresponding to the transmission maxima of the dual-channel FPI. The total SNR, representing the RMS of the SNRs for channels A and B (Eq. (B6)), is also provided. For both receiver channels, reliable wind retrieval is typically achieved when the (total) SNR exceeds a value of 5.





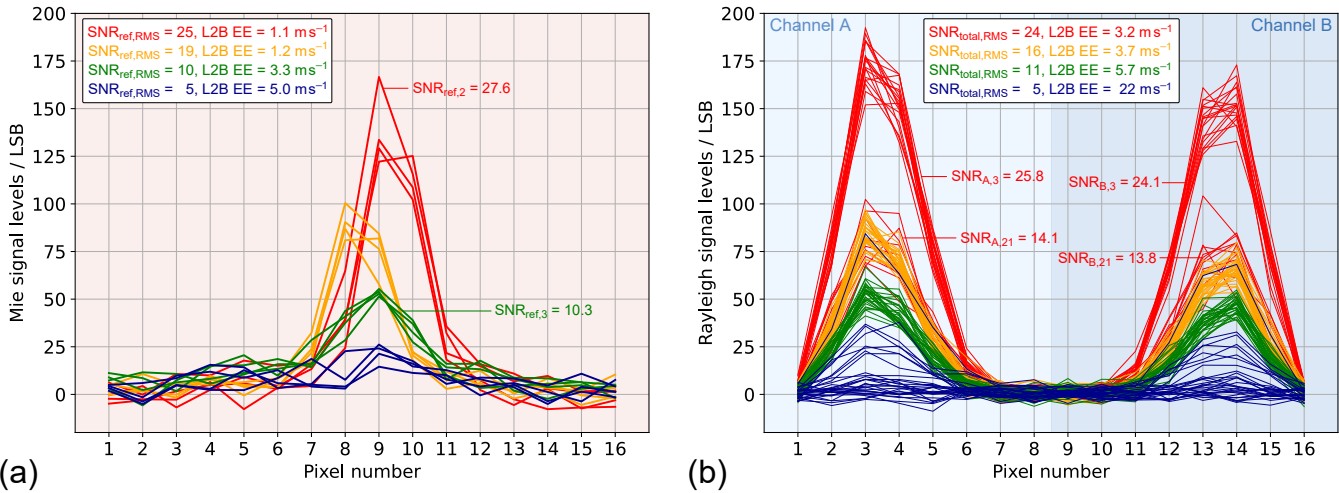

(a)                   (b)

**Figure B1.** (a) Mie and (b) Rayleigh signal levels after DCO-, BKG- and DCMZ-correction for selected measurements from 20 July 2019 between 05:05 UTC and 06:36 UTC (orbit number 5254). Curves with the same colour were grouped to the same L2B Mie-cloudy or L2B Rayleigh-clear wind result, respectively. The different measurement-scale L1B Mie refined SNR and L1B Rayleigh total SNR (RMS over all contributing measurements) are given in the inset along with the L2B EE of the corresponding wind result.

## B4    Evolution of the SNR over the mission

Histograms of Rayleigh total SNR and Mie refined SNR for five selected mission days (Table 1) are displayed in Fig. B2. The SNR values are shown per valid L2B wind result, excluding gross errors identified using the modified Z score method, as

described in Sect. 3.1. To account for varying P/N settings and differing numbers of contributing measurements per wind result (see Appendix A1), the Rayleigh total SNR is normalised to N = 30, i.e., 18 pulses per measurement. When further normalising to a bin thickness of 1 km, the L1B SNR associated with an L2B Rayleigh-clear wind result is calculated as:

$$\mathrm{SNR_{Rayleigh-clear}} = \mathrm{SNR_{Ray,total}}\sqrt{\frac{n}{30}} \cdot \sqrt{\frac{1\,\mathrm{km}}{d_{\mathrm{bin}}}}, \tag{B9}$$

where $n$ is the number of contributing L1B measurements, and $d_{\mathrm{bin}}$ is the vertical bin thickness of the Rayleigh-clear wind.

For Mie-cloudy winds, the L1B Mie refined SNR, calculated as the RMS over all contributing measurements, is scaled only by $\sqrt{n}$. Normalisation to bin thickness is not necessary, as the Mie SNR does not depend on the integration length:

$$\mathrm{SNR_{Mie-cloudy}} = \mathrm{SNR_{Mie,ref}} \cdot \sqrt{n}. \tag{B10}$$

The distribution of the Mie refined SNR remained relatively consistent throughout the mission, characterised by a steep rise from 5 to 10, a peak around $\mathrm{SNR_{Mie,ref}} \approx 10$, and a long tail extending beyond 50. Notably, an SNR threshold of 7 was applied

on measurement level in the L2B processor during the reprocessing of the first FM-A period under Baseline 16, resulting in a




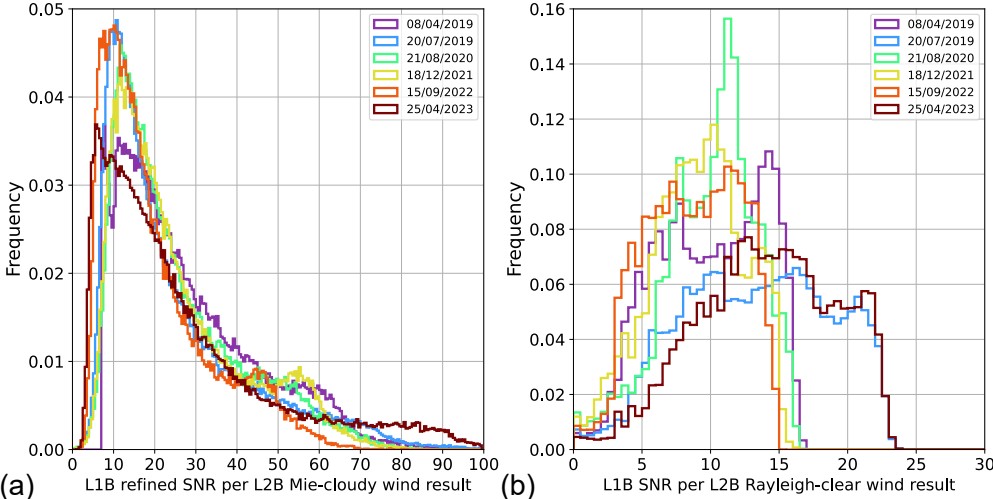

**Figure B2.** Histograms of (a) L1B refined SNR of Mie-cloudy wind results and (b) L1B SNR of Rayleigh-clear wind results for six selected days throughout the mission period. The Rayleigh SNR is normalised to a vertical bin thickness of 1 km and 18 laser pulses per measurement.

sharp cutoff at this value in the dataset from 8 April 2019. However, this setting was found to exclude a substantial number of Mie winds of acceptable quality and was therefore removed in the reprocessing of the remaining mission phases. For Baseline 17, a consistent configuration without an SNR filter is planned to be applied.

The SNR per Rayleigh-clear wind result generally exhibits a narrower distribution than that of the Mie-cloudy winds and shows greater variability over the mission lifetime. The highest values, exceeding 20, were observed during the early FM-B phase in July 2019 and again toward the end of the mission in April 2023. Between 2020 and 2022, the distributions shifted to significantly lower values, ranging from 5 to 15, primarily due to the progressive loss of atmospheric backscatter signal during those years. The median SNR declined from 13.5 in July 2019 to 10.5 in 2020, 9.2 in 2021, and 8.8 in 2022. Following the 860 restoration of emission path transmission through the switch back to the FM-A laser in late 2022, the Rayleigh SNR recovered to 14.2 in 2023.



*Code availability.* The Python code that was used for data analysis and for figure plotting can be provided upon request.

*Data availability.* The presented work includes data of the Aeolus mission which is part of the European Space Agency (ESA) Earth Explorer
Programme. This includes the reprocessed L2B wind product (Baseline 16; de Kloe et al. (2023); https://earth.esa.int/eogateway/documents/
20142/37627/Aeolus-L2B-2C-Input-Output-DD-ICD.pdf, last access: 16 September 2025) that is publicly available and can be accessed via
the ESA Aeolus Online Dissemination System (https://aeolus-ds.eo.esa.int/oads/access/collection/L2B_Wind_Products_Reprocessed, last
access: 26 January 2025; European Space Agency, 2025). The processor development, improvement, and product reprocessing preparation
have been performed by the Aeolus DISC (Data, Innovation and Science Cluster), which involves DLR, DoRIT, TROPOS, ECMWF, KNMI,
CNRS, S&T, ABB, and Serco, in close cooperation with the Aeolus PDGS (Payload Data Ground Segment) from ESA.

*Author contributions.* OL performed the data analysis and wrote the manuscript. MR and JdK developed the L2B processor, remain re-
sponsible for its ongoing development, and contributed to the manuscript. OR supported the interpretation of the results and assisted with
manuscript preparation.

*Competing interests.* The contact author has declared that none of the authors has any competing interests.

*Acknowledgements.* The authors would like to thank Karsten Schmidt from DLR who kindly provided the data for the timeline of the
atmospheric signal levels presented in Fig. 2. Language editing assistance was provided using ChatGPT (OpenAI).



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
