# Peer review of "Five years of Aeolus wind profiling: global coverage and data quality"

_EGUsphere, 2025_

## Referee Comment (RC1)

**Review of "Five years of Aeolus wind profiling: global coverage and data quality" submitted by Lux et al. for publication in EGUsphere**

**Summary of Article & Recommendation**

This paper presents an in-depth quality assessment of the evolution of Aeolus wind data during the first Aeolus mission. The full dataset has been reprocessed using Baseline 16 to provide, for the first time, a consistent dataset covering the entire mission period. The results presented will be used to provide guidance towards the enhancement of data processing and mission/system design for the follow-on mission Aeolus-2, which is currently in development. A significant result of this work is the error decomposition for both Mie and Rayleigh wind regimes, which will positively contribute to improved wind uncertainty characterizations in the Aeolus-2 mission.

I think the content is important and will be valuable to Aeolus-2 and other future wind lidar missions. The article text is very well-written with a few minor grammatical errors; I have indicated these and other thoughts/recommendations in my comments below. Once these have been addressed, my recommendation is that this article is accepted.

**General Comment**

1. Is a similar in-depth analysis on the L2A dataset (aerosols) forthcoming? I think such a study would be an important complement to this one.

**Specific Comments**

- 1. Line 55: "averaged L1B data onto the L2B grid": Were the L1B data in fact averaged or were they interpolated to the L2B grid? The abstract states they were interpolated; if so, what scheme did you use? If they were instead averaged, how was this done? Were only valid obs used? Was any quality control applied to the L1B data beforehand?
- 2. Lines 245-248: Can you provide more detail about how exactly the "Mie-cloudy random error is more strongly affected by data processing algorithms and configuration changes than by the signal trend," instead of just stating that it is? Is the Oct-Nov 2019 example referencing Fig. 2, or something else? How does this example support your claim here?
- 3. Lines 249-251 (and Table 1): This seems out of place here, as the rest of this subsection seems to discuss the Aeolus mission in general and not the actual study presented in this article. I suggest starting a new subsection after line 248 and labeling it as "Dataset selection" or something similar. I further suggest the following:
  - a. Delete the subsection title "2.4 Horizontal bin length of the L2B wind results" and instead incorporate its contents into the new "Dataset selection" subsection.
  - b. In line 249, add "in this study" after "selected for analysis" for clarity.
  - c. Similarly, in line 255, add "in this study" after "wind result" for clarity.

- 4. Line 370: The phrase "increases throughout the troposphere" is confusing here as the next sentence also discusses the troposphere but presents a contradictory statement. I assume the next sentence discusses the *free troposphere* above the PBL? If so, I suggest changing the phrase to "increases throughout the PBL."
- 5. Figure 7: Do the statistics in this figure include winds from both the NH and SH, or just the NH? It is not clear in the text nor in the figure caption. I would explicitly state this, otherwise the results can be confusing. For example, the Tropics panel could be assumed to include all winds south of 20°N (including the entire SH), and the Poles panel could be assumed to only represent the Arctic. Further, if the panels include regions in both hemispheres, have you examined each hemisphere separately? Does the wind coverage differ based on season in the poles/storm track regions, and if so, how?
- 6. Figure 13: A note on discussion order of this figure in the text: Why is panel (b) discussed before panel (a)? Convention dictates that (a) should come first; I recommend matching the discussion in the text to the figure panels, where (a) is discussed first, to be consistent with all other figure discussions.
- 7. Table 2: A note on discussion order of this table in the text: Why are the Rayleigh winds discussed before the Mie winds, when the Mie winds appear first in the table? Tables are typically read top-to-bottom; I recommend matching the discussion in the text to the table's top-to-bottom contents, where the top section (Mie) is discussed first, to be consistent with all other figure discussions.

**Technical Corrections**

**Text**

- 1. Line 40: Past tense should be used: Change "vary" to "varied"
- 2. Line 65: To be consistent with predefined acronyms, I suggest replacing "Level-2B" with "L2B"
- 3. Line 66: See previous comment (Line 65).
- 4. Line 77: Define ALADIN here instead of in line 85.
- 5. Line 85: Don't define ALADIN here (rather, define it in line 77).
- 6. Lines 317-319: This is one sentence and as such does not constitute a paragraph. Please move to the end of the previous paragraph.
- 7. Line 350: Is the comma after "coverage" needed?
- 8. Line 373: Replace the semicolon after "troposphere" with a comma.
- 9. *Lines 437-438*: Tense mismatch: Either pluralize "thickness" in line 437 and replace "was" with "were" at the end of line 438, or replace "were" with "was" in the beginning of line 438.
- 10. Lines 490-491: This is one sentence and as such does not constitute a paragraph. Please move to the end of the previous paragraph.
- 11. Lines 560-561: It looks like the July 2019 and April 2023 tails extend to 12 m/s. Therefore, I would either say "extends up to 12" or "exceeds 10" here.
- 12. Line 561: The low-SNR datasets well exceed 12 m/s in Fig. 13b. Please update the stated range accordingly.
- 13. Line 648: Delete the extra space before "70 %".
- 14. Line 690: HLOS has already been defined in the main text. No need to redefine it here.
- 15. Line 783: DCO has already been defined in the main text. No need to redefine it here.
- 16. Line 838: I believe you mean six selected mission days? Please revise.

**[Technical Corrections cont.]**

**Figures**

- 1. Figure 5 caption: Please pluralize "red" and "blue," as different shades of each are displayed.
- 2. Figure 7: The red and brown dashed lines are difficult to see in panels (b) and (c). And what I assume is the red line for clouds (that which somewhat outlines the Mie wind bars) is too close in color to the dark blue line representing molecules. Please replot with more distinct color choices for the dashed lines.
- 3. *Figure 15:* In the caption, you state that the grey horizontal lines indicate Aeolus mission requirements for two regions in the vertical. However, the grey horizontal lines I see in this figure are those that correspond to the y-axis grid tick marks. Are you perhaps referring to the lines labeled 2.5 m/s and 1.0 m/s? If so, please explicitly state that they are labeled as such in the caption. If not, please add to the panels the Aeolus mission requirement lines in some color other than grey.

---

## Author Comment (AC1)

**Response to Referee Comment (RC1) on**

Five years of Aeolus wind profiling: global coverage and data quality (https://doi.org/10.5194/egusphere-2025-4596)

We sincerely appreciate the referee's insightful and detailed comments on our manuscript. Below, we provide responses to each comment along with the corresponding changes made to the manuscript.

**General comment:**

Is a similar in-depth analysis on the L2A dataset (aerosols) forthcoming? I think such a study would be an important complement to this one.

**Response to General Comment:**

Thank you for this comment. We fully agree that a similar study on the Aeolus L2A aerosol product would be useful and relevant for the wider scientific community using these data. Currently, no preparations are ongoing for such an analysis. However, several publications address the Aeolus L2A product and the algorithms applied in the L2A processor, for example:

Wang, P., Donovan, D. P., van Zadelhoff, G.-J., de Kloe, J., Huber, D., and Reissig, K.: Evaluation of Aeolus feature mask and particle extinction coefficient profile products using CALIPSO data, Atmos. Meas. Tech., 17, 5935–5955, https://doi.org/10.5194/amt-17-5935-2024

Ehlers, F., Flament, T., Dabas, A., Trapon, D., Lacour, A., Baars, H., and Straume-Lindner, A. G.: Optimization of Aeolus' aerosol optical properties by maximum-likelihood estimation, Atmos. Meas. Tech., 15, 185–203, <a href="https://doi.org/10.5194/amt-15-185-2022">https://doi.org/10.5194/amt-15-185-2022</a>

Flament, T., Trapon, D., Lacour, A., Dabas, A., Ehlers, F., and Huber, D.: Aeolus L2A aerosol optical properties product: standard correct algorithm and Mie correct algorithm, Atmos. Meas. Tech., 14, 7851–7871, https://doi.org/10.5194/amt-14-7851-2021.

The co-authors of the present work are primarily focused on the L2B product and do not currently intend to publish a similar study on the L2A product. Nevertheless, we will pass this suggestion on to the L2A experts within the Aeolus Data Innovation and Science Cluster (DISC) to help promote a forthcoming publication on the aerosol product.

**Specific comment #1:**

Line 55: "averaged L1B data onto the L2B grid": Were the L1B data in fact averaged or were they interpolated to the L2B grid? The abstract states they were interpolated; if so, what scheme did you use? If they were instead averaged, how was this done? Were only valid obs used? Was any quality control applied to the L1B data beforehand?

**Response to Specific comment #1:**

The L1B data, primarily the Rayleigh and Mie (refined) SNR analysed in this study, were indeed averaged, not interpolated. Specifically, the RMS value was calculated over all L1B measurements that contribute to a single L2B wind result within the grouping algorithm. The procedure and the rationale for using the RMS rather than an arithmetic mean are explained in Sect. 2.2.3, while details of the grouping algorithm are provided in Appendices A1 and A2.

A large number of validity flags within the L2B processor determine whether a wind result composed of L1B measurements is flagged as valid or invalid and thus serve as quality control. For example, configurable thresholds are applied to the Mie fringe fit parameters to prevent gross errors in the Mie-cloudy wind product arising from noisy Mie signals. For brevity, we do not describe the full concept of these validity flags in the main text. However, we have added the following sentence at the end of Appendix A2:

"Quality control is performed using numerous validity flags that determine whether a wind result derived from L1B measurements is valid or invalid. For example, configurable thresholds on the Mie fringe fit parameters prevent gross errors in the Miecloudy wind product caused by noisy Mie signals."

To avoid confusion regarding the procedure, we have corrected the term in the abstract to read:

"[...] L1B instrument parameters are averaged onto the L2B wind grid [...]"

**Specific comment #2:**

Lines 245-248: Can you provide more detail about how exactly the "Mie-cloudy random error is more strongly affected by data processing algorithms and configuration changes than by the signal trend," instead of just stating that it is? Is the Oct-Nov 2019 example referencing Fig. 2, or something else? How does this example support your claim here?

**Response to Specific comment #2:**

We have elaborated on this point in the text as follows:

"In contrast, the evolution of the Mie-cloudy random error varied very little during the mission, as it was largely independent of the signal trend under the nominal instrument configuration applied during most of the mission lifetime. However, changes in the data processing algorithms and instrument configurations were observed to influence the Mie random error. One notable example is the application of a dedicated range bin setting in October and November 2019, designed to investigate the correspondence between Aeolus

observations and wind measurements derived from Atmospheric Motion Vectors (AMVs). For this purpose, the vertical thickness of the Mie wind bins was reduced to 250 m within the lowermost 2 km of the atmosphere, enabling higher vertical resolution in the planetary boundary layer, where most AMVs are typically found. The narrower range bins significantly decreased the SNR for most of the retrieved Mie-cloudy wind results, causing Poisson noise to become a noticeable contributor to the random error and increasing it from 3.2 to 3.6 m s-1, as shown in Fig. 2. Another example is the change in the number of accumulated laser pulses implemented in December 2021, which slightly improved the Mie-cloudy random error due to the increased horizontal bin length, as discussed in the next section."

**Specific comment #3:**

Lines 249-251 (and Table 1): This seems out of place here, as the rest of this subsection seems to discuss the Aeolus mission in general and not the actual study presented in this article. I suggest starting a new subsection after line 248 and labeling it as "Dataset selection" or something similar. I further suggest the following:

- a. Delete the subsection title "2.4 Horizontal bin length of the L2B wind results" and instead incorporate its contents into the new "Dataset selection" subsection.
- b. In line 249, add "in this study" after "selected for analysis" for clarity.
- c. Similarly, in line 255, add "in this study" after "wind result" for clarity.

**Response to Specific comment #3:**

We followed the referee's suggestion and introduced a new subsection titled "Dataset selection" after line 248, incorporating the content from the subsection "Horizontal bin length of the L2B wind results." The suggested additions "in this study" were also included in lines 249 and 255 for clarity. Furthermore, the description of the paper structure was updated as follows:

"[...] Section 2.3 presents an overview of instrument performance over the mission lifetime, while Sect. 2.4 introduces the datasets selected for this study and discusses the different horizontal integration lengths applied throughout the mission."

**Specific comment #4:**

Line 370: The phrase "increases throughout the troposphere" is confusing here as the next sentence also discusses the troposphere but presents a contradictory statement. I assume the next sentence discusses the free troposphere above the PBL? If so, I suggest changing the phrase to "increases throughout the PBL."

**Response to Specific comment #4:**

We agree that the original sentences were misleading, as they referred to the coverage increase when viewed from the top to the bottom of the plot in Fig. 6(a). This section has been revised to clarify this point and to explicitly state the two reasons for the decrease in coverage at low altitudes:

"For Mie-cloudy winds, coverage gradually increases from the lower stratosphere down through the troposphere, reaching a maximum between 1 and 2 km. The apparent decrease in coverage below 1 km is caused by signal attenuation from clouds and by the exclusion of ground returns, which are flagged as invalid wind data in the L2B product but are still included in the reference area used to calculate wind data coverage."

**Specific comment #5:**

Figure 7: Do the statistics in this figure include winds from both the NH and SH, or just the NH? It is not clear in the text nor in the figure caption. I would explicitly state this, otherwise the results can be confusing. For example, the Tropics panel could be assumed to include all winds south of 20°N (including the entire SH), and the Poles panel could be assumed to only represent the Arctic. Further, if the panels include regions in both hemispheres, have you examined each hemisphere separately? Does the wind coverage differ based on season in the poles/storm track regions, and if so, how?

**Response to Specific comment #5:**

The statistics include winds from both hemispheres, i.e., from 20°N to 20°S for the tropics and combined winds from 60°N–90°N and 60°S–90°S for the poles. This has been clarified in both the main text and the caption of Fig. 7 as follows:

"The coverage of Rayleigh-clear and Mie-cloudy wind observations from the best-case scenario in July 2019 is analysed across three latitude bands (combined for both hemispheres) and depicted in Fig. 7: (a) the polar region (latitudes above 60°), (b) the tropics (latitudes below 20°), and (c) the storm-track region (latitudes between 40° and 60°)."

and:

"Aeolus Mie-cloudy (reds) and Rayleigh-clear (blues) global wind data coverage on 20 July 2019 in different geographical regions: (a) poles (latitude > 60°N/S); (b) tropics (latitude

Aeolus Mie-cloudy (reds) and Rayleigh-clear (blues) global wind data coverage on 20 July 2019 in different geographical regions: (a) NH polar region (latitude >  $60^{\circ}$ N); (b) NH tropics ( $0^{\circ}$  < latitude <  $20^{\circ}$ N); (c) NH storm track region ( $40^{\circ}$ N < latitude <  $60^{\circ}$ N); (d) SH polar region (latitude >  $60^{\circ}$ S); (e) SH tropics ( $0^{\circ}$  < latitude <  $20^{\circ}$ S); (f) SH storm track region ( $40^{\circ}$ S < latitude <  $60^{\circ}$ S). The colour shading indicates the proportion of data within specific intervals of the absolute value of (O-B) wind speed difference  $\varepsilon$ . The Rayleigh-clear wind error is normalised to a vertical bin thickness of 1 km.

We have added this figure and its description to a new Appendix section (Appendix C).

Regarding seasonality:

Seasonal variations in data coverage are difficult to separate from other influencing factors such as the progressive signal loss during the mission and major events such as the 2019 wildfires and the 2022 Hunga Tonga eruption. Therefore, considering the already substantial length of the paper, we decided not to further elaborate on the seasonality of the coverage.

The following text has been added to the end of Sect. 3.1.1:

"It should be noted that seasonal variations in data coverage are difficult to disentangle from other influencing factors such as the progressive signal loss during the mission and special events like the 2019 wildfires and the 2022 Hunga Tonga eruption. Therefore, seasonality will not be further discussed in this study."

**Specific comment #6:**

Figure 13: A note on discussion order of this figure in the text: Why is panel (b) discussed before panel (a)? Convention dictates that (a) should come first; I recommend matching the discussion in the text to the figure panels, where (a) is discussed first, to be consistent with all other figure discussions.

**Response to Specific comment #6:**

We have revised the discussion of Fig. 13 to match the figure panel order, discussing panel (a) first (Mie-cloudy winds), followed by panel (b) (Rayleigh-clear winds), as follows:

"Histograms of the EE per Mie-cloudy and Rayleigh-clear wind result for the six datasets are shown in Fig. 13. Mie-cloudy EE distributions (panel (a)) are narrower, typically extending only to about 10 m s-1. Excluding the dataset from 15 September 2022, these distributions vary little across the mission, indicating the stable quality of Mie-cloudy winds despite the significant signal loss between 2019 and 2022. The September 2022 dataset stands out with a 2 m s-1 shift in Mie EE, linked to the P/N setting change to 114/5 in April 2022 (Rennie and Isaksen, 2024), where the fact that Mie-cloudy winds were formed from a single L1B measurement may have caused artifacts in the fit covariance matrix. Interestingly, Mie EE values in 2023 decreased again despite the same P/N settings and wind grouping, suggesting a processing issue still under investigation for resolution in Baseline 17.

In contrast, as with the Rayleigh SNR, the Rayleigh EE (panel (b)) was normalised to a bin thickness of 1 km. Its distribution features a steep peak and a long tail. For high-SNR datasets (July 2019 and April 2023), the Rayleigh EE peaks around 3 m s-1 and extends up to  $12 \text{ m s}^{-1}$ . As expected from Eq. (3), the EE shifts to larger values for low-SNR datasets, well exceeding  $12 \text{ m s}^{-1}$ . Notably, a small fraction (<1%) of Rayleigh-clear winds show unrealistically high EE values, up to several tens or even  $100 \text{ m s}^{-1}$  (not shown in the plot), which are addressed later in the text."

**Specific comment #7:**

Table 2: A note on discussion order of this table in the text: Why are the Rayleigh winds discussed before the Mie winds, when the Mie winds appear first in the table? Tables are typically read top-to-bottom; I recommend matching the discussion in the text to the table's top-to-bottom contents, where the top section (Mie) is discussed first, to be consistent with all other figure discussions.

**Response to Specific comment #7:**

We have revised the discussion of Table 2 to match the top-to-bottom order of the table, discussing Mie winds first, followed by Rayleigh winds, as follows:

"Finally, Table 2 summarises key statistical parameters describing the Mie and Rayleigh channel performance for the six selected datasets. Mie wind coverage was less sensitive to the loss in atmospheric return signal and more influenced by cloud and aerosol variability, with notable enhancements following events such as the 2019 wildfires and the 2022 Hunga Tonga eruption. The increase in Mie-cloudy wind coverage in April 2023, despite no major aerosol event, suggests that improved signal transmission also benefits the Mie channel. The combination of longer horizontal accumulation (17 km at N=5) and stronger backscatter allowed retrievals from weaker aerosol layers that previously fell below the detection threshold, particularly below  $10 \, \text{km}$ , with a similar increase observed in both hemispheres (Fig. 6).

In contrast, the Rayleigh median SNR dropped from 13.5 in 2019 to 8.8 in 2022 due to the signal loss, resulting in a higher wind random error (from 4.5 to 7.0 m s-1) and similarly increased EE. While overall Rayleigh-clear wind coverage decreased only slightly (from 96 % to 90 %), the share of high-quality winds ( $\varepsilon$  < 2.5 m s-1) declined more noticeably from 74 % to 62 %."

**Technical correction #1:**

Line 40: Past tense should be used: Change "vary" to "varied".

**Response to Technical correction #1:**

The tense has been revised accordingly.

**Technical correction #2:**

Line 65: To be consistent with predefined acronyms, I suggest replacing "Level-2B" with "L2B".

**Response to Technical correction #2:**

The term has been replaced with "L2B" for consistency.

**Technical correction #3:**

Line 66: See previous comment (Line 65).

**Response to Technical correction #3:**

Same correction applied as in Line 65.

**Technical correction #4:**

Line 77: Define ALADIN here instead of in line 85.

**Response to Technical correction #4:**

The definition of ALADIN has been moved to line 77.

**Technical correction #5:**

Line 85: Don't define ALADIN here (rather, define it in line 77).

**Response to Technical correction #5:**

The redundant definition has been removed.

**Technical correction #6:**

Lines 317-319: This is one sentence and as such does not constitute a paragraph. Please move to the end of the previous paragraph.

**Response to Technical correction #6:**

Agreed. The sentence has been moved to the end of the previous paragraph.

**Technical correction #7:**

*Line 350: Is the comma after "coverage" needed?*

**Response to Technical correction #7:**

The comma after "coverage" has been removed. Thank you for noticing.

**Technical correction #8:**

Line 373: Replace the semicolon after "troposphere" with a comma.

**Response to Technical correction #8:**

We have removed the word "however", as it no longer fits with the preceding sentences, and consequently the semicolon was also removed.

**Technical correction #9:**

Lines 437-438: Tense mismatch: Either pluralize "thickness" in line 437 and replace "was" with "were" at the end of line 438, or replace "were" with "was" in the beginning of line 438.

**Response to Technical correction #9:**

Thank you for pointing out this mismatch. We have revised the sentence to:

"The thicknesses of the range bins were configured independently for both channels, typically set to 250 m, 500 m, 1000 m, or 2000 m, and were adjustable along the orbit."

**Technical correction #10:**

Lines 490-491: This is one sentence and as such does not constitute a paragraph. Please move to the end of the previous paragraph.

**Response to Technical correction #10:**

Agreed. The sentence has been moved to the end of the previous paragraph.

**Technical correction #11:**

Lines 560-561: It looks like the July 2019 and April 2023 tails extend to 12 m/s. Therefore, I would either say "extends up to 12" or "exceeds 10" here.

**Response to Technical correction #11:**

**We have revised the sentence to:**

"For high-SNR datasets (July 2019 and April 2023), the Rayleigh EE peaks around  $3 \text{ m s}^{-1}$  and extends up to  $12 \text{ m s}^{-1}$ ."

**Technical correction #12:**

Line 561: The low-SNR datasets well exceed 12 m/s in Fig. 13b. Please update the stated range accordingly.

**Response to Technical correction #12:**

We have revised the sentence to:

"As expected from Eq. (3), the EE shifts to larger values for low-SNR datasets, well exceeding 12 m s-1."

**Technical correction #13:**

*Line 648: Delete the extra space before "70 %".*

**Response to Technical correction #13:**

Thank you for spotting this. The issue was caused by an incorrect LaTeX command. It has been corrected and now reads " $\approx$ 70 %".

**Technical correction #14:**

Line 690: HLOS has already been defined in the main text. No need to redefine it here.

**Response to Technical correction #14:**

The redundant definition has been removed as suggested.

**Technical correction #15:**

Line 783: DCO has already been defined in the main text. No need to redefine it here.

**Response to Technical correction #15:**

The redundant definition has been removed as suggested.

**Technical correction #16:**

Line 838: I believe you mean six selected mission days? Please revise.

**Response to Technical correction #16:**

You are absolutely right. The text has been revised accordingly.

**Technical correction #17:**

Figure 5 caption: Please pluralize "red" and "blue," as different shades of each are displayed.

**Response to Technical correction #17:**

Thank you for the suggestion. The figure caption has been revised to:

"Aeolus Mie-cloudy (reds) and Rayleigh-clear (blues) global wind data coverage for selected days throughout the mission period. [...]"

The same revisions have been applied to the caption of Fig. 7.

**Technical correction #18:**

Figure 7: The red and brown dashed lines are difficult to see in panels (b) and (c). And what I assume is the red line for clouds (that which somewhat outlines the Mie wind bars) is too close in color to the dark blue line representing molecules. Please replot with more distinct color choices for the dashed lines.

**Response to Technical correction #18:**

The colours of the dashed lines have been updated to improve visibility and distinction (see below).

The figure caption has been revised to:

"[...] The dashed lines in panels (b) and (c) represent pre-launch LIPAS simulations of the coverage from clouds (orange), aerosols (green) and molecules (purple), adapted from Marseille et al. (2001)."

**Technical correction #19:**

Figure 15: In the caption, you state that the grey horizontal lines indicate Aeolus mission requirements for two regions in the vertical. However, the grey horizontal lines I see in this figure are those that correspond to the y-axis grid tick marks. Are you perhaps referring to the lines labeled 2.5 m/s and 1.0 m/s? If so, please explicitly state that they are labeled as such in the caption. If not, please add to the panels the Aeolus mission requirement lines in some color other than grey.

**Response to Technical correction #19:**

The horizontal lines indicating the Aeolus mission requirements have been changed to black to distinguish them from the gridlines. The figure caption has been revised to:

"[...] The black dashed horizontal lines indicate the Aeolus mission requirements for the free troposphere (2–16 km) and the PBL (0–2 km), respectively."

---

## Author Comment (AC2)

***Response to Referee Comment (RC2) on***

*Five years of Aeolus wind profiling: global coverage and data quality*

*(*https://doi.org/10.5194/egusphere-2025-4596*)*

We thank the reviewer for the thoughtful and constructive comments on our manuscript. Below, we address each point in detail and describe the revisions made in response.

General comments:

*The purpose of the paper is clear and well stated. However, I - not an Aeolus user- have found the paper difficult to follow in some parts. Sometimes the used terminology and notation does not help the reader; there are some inconsistencies across the paper (e.g. different model background errors are assumed in different figures); it is not clear what are some mean or median values computed from (e.g. Fig. 2 depicts daily average random errors computed across what measurements? All of them? not excluding anything?). Also, the SNR terminology is in my opinion kind of misleading. The SNR is the SNR, if we start scaling it with the square root of the number of measurements is not an SNR any more (of course the latter is the quantity that is relevant to the measurement accuracy, by averaging we are not improving the SNR we are improving the sensitivity). In fact, it is named "scaled SNR" (but again to me the naming is not really helping the reader).*

Response to General Comments:

We thank the referee for the constructive feedback and for recognizing that the purpose of the paper is clearly stated. We also appreciate the careful reading and the detailed comments regarding clarity, terminology, and notation.

We acknowledge that some parts of the manuscript may be difficult to follow for readers who are not familiar with Aeolus data, and that certain terminology, notation, and figure descriptions could be made more transparent. We also note the referee's observations regarding inconsistencies in assumed model background errors and the computation of mean or median values.

We have carefully reviewed the manuscript and will address these points individually in the following responses. We believe that the detailed point-by-point clarifications and revisions described below will resolve the concerns raised and improve the readability of the manuscript for a broader audience.

Regarding the SNR, we could not identify any instance in the manuscript where the term "scaled SNR" is used. In Eqs. (B9) and (B10), the Rayleigh and Mie SNR are multiplied by the square root of the number of measurements. This scaling is necessary to make SNR values from different periods

of the mission comparable, as the number of accumulated measurements varied over time. It should be noted that this scaling does not alter the definition of SNR, but rather allows comparison of instrument performance and associated wind data quality across periods with different accumulation lengths.

Specific comment #1:

*Line 236: To avoid confusion, I would name it normalised atmospheric signal (NAS) because it is normalised to a specific date.*

Response to Specific comment #1:

Following the referee's suggestion, we have renamed the quantity to normalised atmospheric signal (NAS) throughout the manuscript, including the text, Table 1, and Fig. 2.

Specific comment #2:

*Line 247-248: not sure what you are alluding to. I do not see any peculiarity at that time.*

Response to Specific comment #2:

We agree that this point was unclear. We have therefore elaborated on this in the manuscript by explicitly describing the behaviour observed at that time. The revised text now reads:

*"In contrast, the evolution of the Mie-cloudy random error varied very little during the mission, as it was largely independent of the signal trend under the nominal instrument configuration applied during most of the mission lifetime. However, changes in the data processing algorithms and instrument configurations were observed to influence the Mie random error. One notable example is the application of a dedicated range bin setting in October and November 2019, designed to investigate the correspondence between Aeolus observations and wind measurements derived from Atmospheric Motion Vectors (AMVs). For this purpose, the vertical thickness of the Mie wind bins was reduced to 250 m within the lowermost 2 km of the atmosphere, enabling higher vertical resolution in the planetary boundary layer, where most AMVs are typically found. The narrower range bins significantly decreased the SNR for most of the retrieved Mie-cloudy wind results, causing Poisson noise to become a noticeable contributor to the random error and increasing it from 3.2 to 3.6 m s-1, as shown in Fig. 2. Another example is the change in the number of accumulated laser pulses implemented in December 2021, which slightly improved the Mie-cloudy random error due to the increased horizontal bin length, as discussed in the next section."*

Specific comment #3:

*Fig:3: really difficult to grasp the significance of this figure. Also, not clear to me what is the meaning of the different labels with the different h. Simplify the figure or even delete?*

Response to Specific comment #3:

We thank the referee for this comment and acknowledge that the original presentation of Fig. 3 was not sufficiently clear. We have therefore revised the description in the manuscript to better explain the purpose and interpretation of the figure.

Panels (a) and (b) of Fig. 3 show the cumulative distribution of the number of contributing L1B measurements for L2B Mie-cloudy and Rayleigh-clear wind results, respectively. As the figure combines data from six different days, plotting the cumulative frequency was chosen to avoid strongly overlapping histograms that would be difficult to distinguish in a standard frequency plot.

The stepwise increase in cumulative frequency reflects the discrete number of measurements contributing to each wind result and highlights how different P/N settings used during the mission affect the horizontal accumulation length. We believe this information is important for understanding the representativeness and effective horizontal resolution of the retrieved winds.

The labels indicating different values of *h* refer to the corresponding horizontal bin length implied by the accumulation. This length varies with the P/N setting. For example, accumulating three measurements at $N = 30$ corresponds to a horizontal length of approximately 86 km / 30 × 3 ≈ 9 km, whereas the same number of measurements at $N = 15$ corresponds to approximately 86 km / 15 × 3 ≈ 17 km.

We have clarified this interpretation in the revised text to improve readability and to make the relevance of Fig. 3 clearer to the reader as follows:

> *"Figure 3 shows the number of measurements contributing to each Mie-cloudy (a) and Rayleigh-clear (b) wind result for the six days analysed in this study. This information directly reflects the effective horizontal accumulation length Δx of the retrieved winds and illustrates how changes in the P/N settings during the mission influenced the spatial representativeness of the wind products."*

Specific comment #4:

*Fig.3: h is not a very good name for a horizontal bin length (why not using Dx?). Later on, I see you use d_bin for the vertical bin thickness. In my opinion this is not a good selection of variable names.*

Response to Specific comment #4:

We agree with the referee that the original notation was not ideal. To improve clarity and consistency, we have renamed the horizontal bin length from $h$ to $\Delta x$ and the vertical bin thickness from $d_{\text{bin}}$ to $\Delta y$ throughout the manuscript and figures.

Specific comment #5:

*Eq. (2) not fully clear to me over which dataset the median is computed from (all altitudes?).*

Response to Specific comment #5:

We agree that the formulation of Eq. (2) was not sufficiently clear. Equation (2) introduces the modified Z score in a general form, which is applied in this study to different datasets for outlier detection. For the data coverage analysis described in Sect. 3.1.1 and shown in Fig. 5, the median is computed over all valid wind results within each 1 km altitude bin.

We have revised the text as follows:

> *"The modified Z score is computed as the distance from the median of the regarded dataset, normalised by the scaled median absolute deviation (MAD) [...].*

Specific comment #6:

*Fig.5: I am not sure I see three tones of red. The faintiest red looks more like a white.*

Response to Specific comment #6:

We thank the referee for pointing this out. We have increased the saturation of the faintest red tone in Figs. 5 and 7 so that it is clearly distinguishable from white.

Specific comment #7:

*Fig. 7: why there are no LIPAS simulations in the polar regions?*

Response to Specific comment #7:

The LIPAS simulation curves shown in Fig. 7 are taken from Marseille et al. (2001). That study provides coverage plots only for the tropics and storm-track regions; simulations for polar regions are not included. Consequently, no LIPAS results are available for comparison with the polar data shown in panel (a) of Fig. 7. This limitation is now explicitly stated in the figure caption.

Specific comment #8:

*Line 528 and Fig. 12 "the altitude dependence of the Mie and Rayleigh SNR per wind result". What does it mean? Is this a mean or median value computed across valid data?*

Response to Specific comment #8:

The data basis for Fig. 12 has been clarified in the text as follows:

> *"Figure 12 shows the altitude dependence of the Mie and Rayleigh SNR, calculated as the median SNR of all valid, Z-score-filtered wind results within each 1 km altitude bin."*

Specific comment #9:

*Eq.3: I cannot actually find in App. B a formula for the EE (apart from saying it is inversely proportional to the SNR).*

Response to Specific comment #9:

When introducing the Rayleigh error estimate (EE) in Line 549, the wording was misleading. The reference to Appendix B does not provide an explicit formula for the EE; instead, it defines the Rayleigh SNR, to which the EE is inversely proportional, as expressed in Eq. (3).

We have therefore reformulated the sentence as follows:

> *"The EE scales inversely with the SNR, with the latter defined in Eqs. (B1–B6)."*

As stated earlier in the text, a more detailed description of the computation of the Rayleigh and Mie EE is provided in the Aeolus L2B Algorithm Theoretical Basis Document (Rennie et al., 2020), with a concise overview given in Lux et al. (2022).

Specific comment #10:

*Fig 15: the figure is a little bit messy. I would probably reduce the number of lines to the most significant ones (2 or 3 out of 6). Also, I would expect that the error bars became longer and longer with smaller errors (which seems to occur for panel b but not for panel a).*

Response to Specific comment #10:

We thank the referee for this comment and acknowledge that Fig. 15 may appear visually busy at first glance. We have considered reducing the number of datasets shown; however, omitting some of the investigated cases would reduce the transparency and representativeness of the comparison between the error estimates (dashed lines) and the random wind error derived from (O–B) statistics

(solid lines). The close overlap of the dashed curves, except for the 2022 dataset, reflects the consistent relationship between the error estimate and SNR across the different mission phases, which we consider an important result.

Regarding the error bars, they do indeed increase in length with decreasing wind error, as expected, because the relative contribution of the model background error becomes larger as the observation error decreases. The impression that this behaviour is not present in panel (a) arises from the fact that several error bars for the 2022 dataset are not shown at high-SNR values. This occurs when the random error derived from (O–B) statistics falls below 2.5 m s$^{-1}$, which only affects the three highest-SNR data points of that dataset.

We have clarified this behaviour in the revised figure caption to avoid confusion:

> "L2B wind errors as a function of L1B SNR for the (a) Mie and (b) Rayleigh channels, [...] The error bars indicate the range of observation error when assuming background errors of 1.5 m s$^{-1}$ and 2.5 m s$^{-1}$. Note that the lower error bars cannot be computed for the three highest-SNR data points of the 2022 dataset, as the random error is smaller than 2.5 m s$^{-1}$. [...]"

Specific comment #11:

*Line 604: "the actual error": I am not sure we can refer to that as the actual error. It implies that ECMWF background wind is the "truth" (particularly not applicable for the Mie winds in presence of clouds and convection).*

Response to Specific comment #11:

We agree with the referee that referring to this quantity as the "actual error" is misleading, as it could imply that the ECMWF background wind represents the truth, which is not the case—particularly for Mie-cloudy winds in the presence of clouds and convection. We have therefore revised the wording accordingly. The revised text now reads:

> "For Mie-cloudy winds, the EE slightly underestimates the random error in terms of the scaled MAD of the (O–B) wind speed difference, particularly in the low-error (i.e. high-SNR) regime."

*Line 609-610: not sure I understand this extrapolation. I thought the background error (assumed 2 m/s) should be already subtracted from the y-axis values. Maybe it is worth properly defining epsilon somewhere. In fact, how does the extrapolation work when comparing Rayleigh and Mie winds? Actually, I see some explanation later but does this actually mean that the background error for the Mie winds should be taken much larger than 2 m/s or otherwise that the EE is underestimated at high SNR? I would actually rephrase line 610 accordingly.*

Response to Specific comment #12:

We thank the referee for this detailed comment, which helped to clarify an important conceptual point in the manuscript.

The extrapolation discussed in Lines 609–610 is indeed performed on the wind random error after subtraction of the assumed model background error of 2 m s$^{-1}$. The parameter $\varepsilon$ accounts only for the model background error and does not include the representativeness error. These two contributions are distinct and add quadratically to the (O–B) wind speed difference.

For the model background error, values of $(2.0\pm0.5)$ m s$^{-1}$ are considered reasonable based on Desroziers diagnostics (Desroziers et al., 2005) applied to radiosonde zonal wind and Aeolus HLOS wind departures (Rennie et al., 2021). In contrast, the representativeness error is subject to substantially larger uncertainty and is therefore not included in the definition of $\varepsilon$. For Rayleigh-clear winds, this contribution is negligible due to their typically larger observation error and longer horizontal accumulation length. For Mie-cloudy winds, however, the representativeness error becomes relevant in the high-SNR regime, which is reflected by the increasing deviation between the wind random error and the error estimate (EE) at low EE values in Fig. 15.

The extrapolation towards EE = 0 m s$^{-1}$ is associated with considerable uncertainty, as indicated by the error bars, which represent the uncertainty in the assumed background error. While the results suggest that a larger combined background and representativeness error may be appropriate for Mie-cloudy winds at high SNR, we have refrained from treating Rayleigh and Mie winds differently in this study in order to maintain a consistent methodology. Nevertheless, we consider the increasing relevance of the representativeness error in the high-SNR regime to be an important outcome of this analysis.

To address the referee's concern regarding the definition of $\varepsilon$ and to avoid ambiguity, we have revised the corresponding passage in Sect. 3.1 as detailed below:

*"In the next step, the wind data are classified according to the magnitude of the HLOS wind speed departures from the ECMWF model background (short-range forecast). To this end, the absolute (O–B) departure, ε, is evaluated while accounting for the contribution of the model background error σB, which is treated here as a simplified estimate and does not explicitly represent the scene-dependent variability of background forecast errors σB:*

$$\varepsilon = \sqrt{(v_\mathrm{O} - v_\mathrm{B})^2 - \sigma_\mathrm{B}^2}$$

*The typical range of σB in HLOS wind space is 1.5–2.5 m s⁻¹, as estimated from radiosonde zonal wind and Aeolus HLOS wind departures using Desroziers diagnostics (Desroziers et al., 2005; Rennie et al., 2021). Owing to the variability of σB with altitude and geolocation, this range is used to define confidence intervals for the Aeolus random wind errors shown in Fig. 2. For the classification of Aeolus wind data quality using ε, the upper value of the background error range (σB = 2.5 m s⁻¹) is assumed, representing a best-case scenario for Aeolus data coverage. Wind results with ε < 2.5 m s⁻¹ are classified as 'high-quality', those with 2.5 m s⁻¹ < ε < 5.0 m s⁻¹ as 'medium-quality', and those with ε > 5.0 m s⁻¹ as 'low-quality'."*

Specific comment #13:

*Figure A1: I have to say that I struggle understanding the meaning of the last box "Dimensions". It is not very intuitive for somebody who is not an Aeolus user to figure out what is meant with observation, measurement, pixel, wind result (#profiles #range bins, # averaged profiles, ... are simpler concepts to grasp).*

Response to Specific comment #13:

We appreciate the referee's comment and acknowledge that the terminology used in Fig. A1 is specific to the Aeolus Level-1 and Level-2 data products. The terms "observation", "measurement", "pixel", and "wind result" follow the established nomenclature used in the Aeolus Algorithm Theoretical Basis Documents, which are referenced in the manuscript, and are also introduced in Sect. 2.2 describing the Aeolus data structure (see lines 133 ff.). For the sake of consistency with the main text and the official Aeolus documentation, we therefore retain this terminology in Fig. A1.

To improve the accessibility of the figure for readers who are not familiar with Aeolus, we have revised the figure caption to more explicitly explain the meaning of the "Dimensions" box:

> *"Overview of the L1A, L1B, and L2B product parameters analysed in this study, together with their respective dimensions. The dimensions indicate the number of elements at each processing level, e.g. measurements and observations (horizontal scale), range bins (vertical scale), and retrieved wind results (including horizontal and vertical scales)."*

Specific comment #14:

*Figure A2 is also difficult. A simple schematic with the terminology used could help.*

Response to Specific comment #14:

We acknowledge that Fig. A2 may be difficult to follow, especially for readers who are not familiar with the Aeolus Level-2B processing. The grouping algorithm implemented in the Level-2B processor is inherently complex, and the figure aims to illustrate the key concepts of the transition from the two-dimensional L1B data structure to the one-dimensional L2B wind product, as well as the distinction between cloudy and clear wind results.

To improve the accessibility of the figure, we have revised the caption to more clearly describe the processing steps and terminology illustrated in the schematic:

> *"Schematic illustrating the grouping algorithm for (a) Mie winds and (b) Rayleigh winds in the Aeolus L2B processor. Individual L1B measurements, per range bin and observation, are classified as "cloudy" or "clear" and then gathered into groups according to the mission parameter settings. These groups are alternately stacked into a one-dimensional array in the L2B product, with L2B wind results enumerated following the order of measurements within each observation. The example shown is based on L1B and L2B data from orbit 21857 (1 June 2022, 05:49–07:19 UTC) with P/N settings 114/5. For other P/N settings, multiple L1B measurements may contribute to a single L2B Mie wind result. Due to the longer accumulation length of Rayleigh winds, multiple L1B measurements contribute to a single Rayleigh wind result even for P/N 114/5, as exemplified for wind result 2 (Rayleigh-cloudy) and wind result 5 (Rayleigh-clear)."*

As Fig. A2 is included in the Appendix, it is intended as supplementary material for readers who wish to gain additional insight into the Aeolus processing chain, without having to consult the much more detailed Algorithm Theoretical Basis Documents, where the grouping algorithm is described over several pages.

Specific comment #15:

*Line 613: "yielding unrealistically high values", I am not sure this are unrealistic values for the errors, to me they may be very realistic (rephrase?)*

Response to Specific comment #15:

We thank the referee for this comment and agree that the wording could be interpreted too strongly. The statement refers specifically to the range of Rayleigh error estimates (EE) obtained when the SNR falls below 5, which predominantly occurs beneath optically thick clouds. In this regime, the EE increases to several tens of m s$^{-1}$ and can exceed 100 m s$^{-1}$. While such values are not unphysical in a strict sense, they are considered too large for the wind products to be meaningful or reliable for scientific or practical use.

We have therefore retained the statement but clarified the context and magnitude of the error values being referred to. The revised text now reads:

> *"As shown in Fig. 15(b) and consistent with Eq. (4), the EE diverges when the SNR drops below 5, yielding very large error estimates (often several tens of m s$^{-1}$ or even exceeding 100 m s$^{-1}$). These low-SNR winds, typically retrieved beneath optically thick clouds (see Fig. 11), are not considered reliable."*

Specific comment #16:

*model background error: sometimes it is assumed 2.0 m/s sometimes 2.5 m/s. I would try to keep consistency across the paper.*

Response to Specific comment #16:

We agree with the referee that consistency in the assumed model background error is important. As discussed in our response to comment #12, the model background error is estimated to lie in the range 1.5–2.5 m s$^{-1}$. Throughout most of the manuscript, we therefore use the mean value of 2.0 m s$^{-1}$. An exception is made in Sect. 3.1 for the classification of high- and medium-quality winds, where a value of 2.5 m s$^{-1}$ is adopted to represent a best-case scenario for Aeolus data coverage.

To assess the impact of this choice, we repeated the classification using a background error of 2.0 m s$^{-1}$. The resulting data coverage for high- and medium-quality winds decreases by less than 5 %. For example, for the 20/07/2019 dataset listed in Table 2, the coverage of high-quality Rayleigh-clear winds is reduced from 74 % to 72 %, and that of high-quality Mie-cloudy winds from 9.1 % to 8.9 %.

We have clarified this choice and its impact in the revised manuscript text as follows:

> *"Note that a model background error of 2.5 m s$^{-1}$ is assumed for the classification of high- and medium-quality winds to represent a best-case scenario for Aeolus data coverage, whereas a value of 2.0 m s$^{-1}$ is used elsewhere in the study. Using 2.0 m s$^{-1}$ for the classification would lead to a slight reduction (less than 5 %) in the estimated data coverage. For example, for the 20/07/2019 dataset listed in Table 2, the coverage of high-quality Mie-cloudy and Rayleigh-clear winds would decrease from 9.1 % to 8.9 % and from 74 % to 72 % respectively."*

Specific comment #17:

*Line 677: I would recommend to introduce also the reference Illingworth, A. J., and Coauthors, 2018: WIVERN: A New Satellite Concept to Provide Global In-Cloud Winds, Precipitation, and Cloud Properties. Bull. Amer. Meteor. Soc., 99, 1669–1687, https://doi.org/10.1175/BAMS-D-16-0047.1.*

Response to Specific comment #17:

We thank the referee for this suggestion. The reference to Illingworth et al. (2018) has been added at the appropriate place in the manuscript.

Technical correction #1:

*Lin373: ; ==> ,*

Response to Technical correction #1:

We have rephrased this sentence as follows:

> *"The apparent decrease in coverage below 1 km is caused by signal attenuation from clouds and by the exclusion of ground returns, which are flagged as invalid wind data in the L2B product but are still included in the reference area used to calculate wind data coverage. Within the troposphere coverage shows significant variability among the six selected days influenced by the presence of clouds, without any consistent trend over the mission duration."*